# A Comprehensive Information-Decomposition Analysis of Large Vision-Language Models

**Lixin Xiu[1], Xufang Luo[2], Hideki Nakayama[1]**
[1]The University of Tokyo
[2]Microsoft Research

## Abstract

Large vision-language models (LVLMs) achieve impressive performance, yet their internal decision-making processes remain opaque, making it difficult to determine if the success stems from true multimodal fusion or from reliance on unimodal priors. To address this attribution gap, we introduce a novel framework using partial information decomposition (PID) to quantitatively measure the "information spectrum" of LVLMs—decomposing a model's decision-relevant information into redundant, unique, and synergistic components. By adapting a scalable estimator to modern LVLM outputs, our model-agnostic pipeline profiles 26 LVLMs on four datasets across three dimensions—*breadth* (cross-model & cross-task), *depth* (layer-wise information dynamics), and *time* (learning dynamics across training). Our analysis reveals two key results: (i) two task regimes (synergy-driven vs. knowledge-driven) and (ii) two stable, contrasting family-level strategies (fusion-centric vs. language-centric). We also uncover a consistent three-phase pattern in layer-wise processing and identify visual instruction tuning as the key stage where fusion is learned. Together, these contributions provide a quantitative lens beyond accuracy-only evaluation and offer insights for analyzing and designing the next generation of LVLMs. Code and data are available at https://github.com/RiiShin/pid-lvlm-analysis.

## 1 Introduction

Large vision-language models (LVLMs) achieve remarkable success across a wide range of multimodal tasks, including visual question answering (Chen et al., 2024), image captioning (Bai et al., 2025a), and open-ended reasoning (Zhu et al., 2025a). However, the internal mechanisms driving these impressive results remain largely opaque. Accuracy and other aggregate performance metrics only reflect the final outcomes of model predictions, but not the underlying processes through which these outcomes are obtained. While prior research has begun to analyze large language models (LLMs) to isolate factors shaping predictions (Jain & Wallace, 2019; Meng et al., 2022), LVLMs pose a distinct challenge because they must process and integrate multiple modalities. In particular, understanding whether a model's prediction is primarily driven by visual evidence, language priors, or the interaction between the two is critical for interpreting its behavior. However, existing interpretability efforts often adopt a "micro-scope" focus—analyzing one modality in isolation—or introduce ad hoc metrics that lack firm theoretical support. Consequently, the field still lacks comprehensive, quantitative tools capable of dissecting the internal strategies by which LVLMs use multimodal information.

To address this challenge, we introduce a framework built on partial information decomposition (PID) to comprehensively and quantitatively analyze LVLM behavior. PID is a rigorous information-theoretic methodology that decomposes the mutual information between a set of inputs and an output into distinct components. Originally developed in neuroscience and complex systems (Williams & Beer, 2010), PID characterizes how multiple information channels jointly influence a target variable. We extend this perspective to LVLM inference by treating vision features $X_1$ and language features $X_2$ as inputs and the model prediction $Y$ as the output, partitioning decision-relevant informa-

---

Correspondence to: Xufang Luo <xufluo@microsoft.com> and Hideki Nakayama <nakayama@ci.i.u-tokyo.ac.jp>.

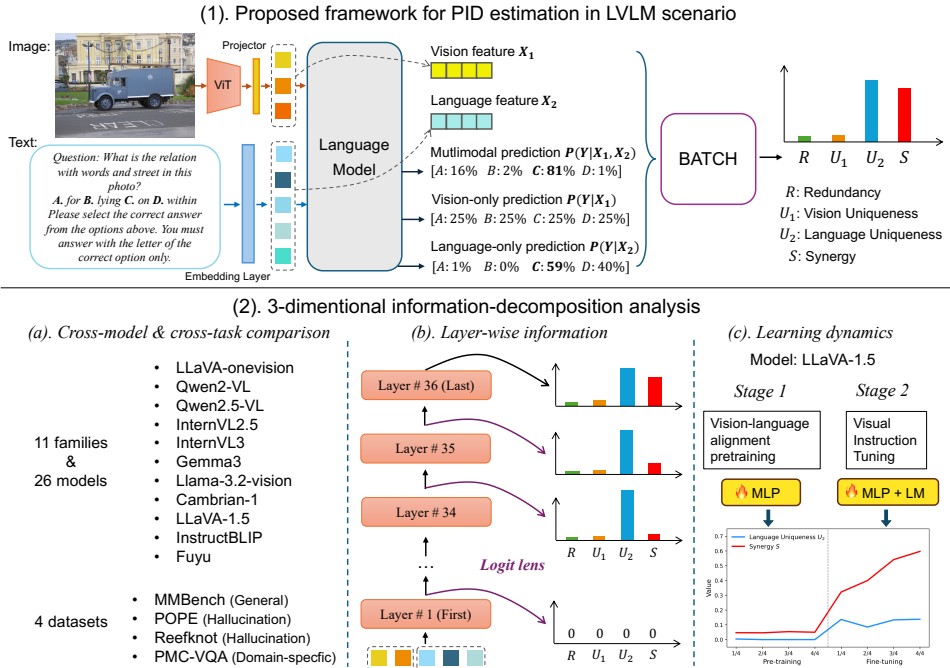

Figure 1: Overview of this research. The first part is the framework of PID estimation for LVLMs. Given an image-text pair, we extract image and text embeddings as two features, run a standard multimodal forward pass and collect two unimodal predictions by masking the other modality. PID values are estimated with BATCH estimator. The second part reveals three analysis dimensions: (1) cross-model and cross-task comparison, (2) layer-wise information dynamics, and (3) learning dynamics over training. To our knowledge, this is the first comprehensive LVLM analysis through the lens of information decomposition.

tion into four non-negative terms: redundancy $R$ (shared by both), vision uniqueness $U_1$, language uniqueness $U_2$, and synergy $S$ (emerging only from their combination). We refer to $\{R, U_1, U_2, S\}$ as the model's *information spectrum*. It provides a principled lens for probing LVLM internals and, unlike "micro-scope" analyses or more empirical approaches, enables a quantitative separation of the model's core information-processing strategies.

To apply PID to modern LVLMs, we adapt the PID estimator proposed by Liang et al. (2023a) for visual question-answering (VQA) tasks and propose a model-agnostic pipeline that requires no architectural changes or retraining. Using this pipeline, we analyze LVLMs along three axes: (1) a cross-model, cross-task comparison spanning 26 models and four benchmarks; (2) layer-wise information dynamics via a logit-lens view; and (3) the evolution of multimodal fusion across training stages. The full framework and analysis dimensions are summarized in Figure 1.

Our study yields several actionable insights. We identify two task regimes—*synergy-driven* vs. *knowledge-driven*—and show that model families themselves adopt two stable, contrasting strategies—*fusion-centric* vs. *language-centric*. We clarify how fusion develops, observing a three-phase pattern over layers and finding that visual instruction tuning is the key stage where $S$ is unlocked. Taken together, these results provide a quantitative basis for moving beyond accuracy-only evaluation toward a more principled understanding of multimodal processing in LVLMs.

## 2 RELATED WORK

**The evolution of vision-language models.** Vision-language models (VLMs) have shifted from contrastive dual-encoder paradigms to generative paradigms. While early dual-encoder models like CLIP (Radford et al., 2021) and ALIGN (Jia et al., 2021) focused on joint representation learning, the current approach uses a parameter-efficient generative architecture, typically comprising a vi-

sion encoder, a projector, and a large language model (LLM) backbone (Tsimpoukelli et al., 2021; Alayrac et al., 2022; Merullo et al., 2022; Li et al., 2023a).

The evolution of large vision-language models (LVLMs) is driven by advances in LLM backbones and training methodologies (Dai et al., 2024). Architecturally, backbones have shifted from encoder-decoder models like T5 (Raffel et al., 2020) to decoder-only models such as Llama (Touvron et al., 2023). Methodologically, performance has been greatly enhanced by various training stages on vast, high-quality datasets, a strategy pioneered by LLaVA (Liu et al., 2023b) and MiniGPT-4 (Zhu et al., 2023). State-of-the-art LVLMs integrate the most powerful backbones (e.g., Llama3.1 (Grattafiori et al., 2024) and Qwen2.5 (Qwen et al., 2025)) with advanced training recipes (Liu et al., 2023a; Li et al., 2024; Wang et al., 2024; Chen et al., 2024; Tong et al., 2024; Meta AI, 2024; Bai et al., 2025a; Zhu et al., 2025a), resulting in more capable yet more complex systems.

**Probing the black box: interpretability in VLMs.**    To address the opacity of VLMs, researchers have adapted interpretability techniques from transformer-based language models (Vaswani et al., 2017). One line of work generates post-hoc explanations to identify influential inputs using attribution heatmaps (Schulz et al., 2020; Wang et al., 2023), attention maps (Abnar & Zuidema, 2020; Chefer et al., 2021; Gandelsman et al., 2024), and activation analysis (Conmy et al., 2023; Arditi et al., 2024). Complementary approaches probe a model's internal representations to understand its learned knowledge (Meng et al., 2022), for instance by training linear probes (Alain & Bengio, 2016) to decode features in LLMs (Hewitt & Manning, 2019; Tenney et al., 2019), applying the logit lens (nostalgebraist, 2020; Belrose et al., 2023) to inspect LLMs' intermediate computations (Cywiński et al., 2025; Neo et al., 2024), or identifying "multimodal neurons" corresponding to human-interpretable concepts (Goh et al., 2021; Schwettmann et al., 2023).

With the rise of LVLMs, research is shifting toward understanding their deep multimodal fusion. Recent work in this area includes analyzing and manipulating the visual token representations that bridge the two modalities (Jiang et al., 2024b; Basu et al., 2024; Liu et al., 2025), as well as quantifying vision's contribution through analyses of visual attention sinks (Kang et al., 2025) and cross-modal information-flow tracing (Zhang et al., 2025; Nikankin et al., 2025; Yang et al., 2024).

**An information-theoretic lens on multimodal learning.**    Information theory offers a quantitative framework for analyzing information flow in neural networks. Foundational concepts like mutual information (MI) (Shannon, 1948) and the subsequent information bottleneck (IB) principle (Tishby et al., 2000) have been widely used in multimodal learning. Applications range from using MI for interpretability (Oh et al., 2025) to employing the IB framework for both guiding representation learning (Almudévar et al., 2025; Jiang et al., 2024a; Xiao et al., 2024; Wu et al., 2025; Bai et al., 2025b) and enhancing model transparency (Wang et al., 2023; Zhu et al., 2025b).

While mutual information can quantify the total information from a source, it cannot disentangle the complex interactions between multiple inputs, such as vision and text. To address this, partial information decomposition (PID) (Williams & Beer, 2010) decomposes the information about a target variable into redundant, unique, and synergistic components. The application of PID to machine learning is a nascent field (Ehrlich et al., 2022; Dissanayake et al., 2025; Choi et al., 2025; Shan et al., 2025), though recent studies have begun to apply it within multimodal learning contexts (Liang et al., 2023a;b; 2024). However, its use for analyzing the composition, flow, and evolution of multimodal information within modern LVLMs remains unexplored, which is a gap this paper aims to fill.

## 3    METHODOLOGY

This section details our methodology for applying PID to analyze LVLMs in three parts. We first review the fundamentals of PID and the specific estimator our work adapts in Section 3.1. We then present our framework, including three key adaptations that make this estimator robust for the unique context of modern LVLMs, in Section 3.2. Finally, we outline the experimental design that leverages this framework to conduct a comprehensive analysis across three dimensions in Section 3.3.

### 3.1 PRELIMINARIES

**Partial information decomposition.** Mutual information (Shannon, 1948) measures the statistical dependence between two variables. However, in a three-variable system comprising two source variables $X_1, X_2$ and a target variable $Y$, with respective state spaces $\mathcal{X}_1, \mathcal{X}_2$, and $\mathcal{Y}$, the standard interaction information $I(X_1; X_2; Y)$ can be negative, limiting its interpretability.

To address this, partial information decomposition (PID) (Williams & Beer, 2010) reframes the problem by decomposing the total mutual information $I(X_1, X_2; Y)$ into 3 non-negative atoms: **redundancy** (information common to both sources), **uniqueness** (information exclusive to each source), and **synergy** (new information emerging from their combination). Following Bertschinger et al. (2014), the components are defined on the set of distributions $\Delta_P = \big\{ Q \in \Delta : Q(x_i, y) = P(x_i, y), \ \forall x_i \in \mathcal{X}_i, \ y \in \mathcal{Y}, \ i \in \{1, 2\} \big\}$, which contains all joint distributions $Q$ over $(X_1, X_2, Y)$ that preserve the source-target marginals of the true distribution $P$. The atoms are given as:

$$R = \max_{Q \in \Delta_P} I_Q(X_1; X_2; Y), \tag{1}$$

$$U_1 = \min_{Q \in \Delta_P} I_Q(X_1; Y \mid X_2), \tag{2}$$

$$U_2 = \min_{Q \in \Delta_P} I_Q(X_2; Y \mid X_1), \tag{3}$$

$$S = I(X_1, X_2; Y) - \min_{Q \in \Delta_P} I_Q(X_1, X_2; Y). \tag{4}$$

This decomposition provides a principled framework for quantifying how individual and joint sources of information contribute to a target variable, but estimating these atoms from data is a non-trivial task that requires specialized estimators.

**Estimating PID for multimodal inputs.** To estimate PID for the high-dimensional representations within modern LVLMs, we adapt the scalable estimator from Liang et al. (2023a). This work introduces two methods: a convex programming-based estimator CVX for discrete features, and an approximate estimator **BATCH** designed for continuous, high-dimensional modalities.

Our work builds upon the BATCH estimator, as it is well-suited for analyzing the continuous vectorial embeddings produced by LVLMs. This method uses neural networks to parameterize the required probability distributions. It then optimizes an information-theoretic objective over minibatches, employing a variant of the Sinkhorn algorithm (Cuturi, 2013) to enforce the marginal-matching constraints defined in $\Delta_P$.

See Appendix A for more details on PID and the BATCH estimator.

### 3.2 A PID ESTIMATION FRAMEWORK FOR LVLMS

We propose a PID estimation framework for LVLMs tailored to multiple-choice visual question answering (MC-VQA) tasks. This focus is a deliberate methodological choice: BATCH requires a finite $\mathcal{Y}$, thus the natural set of choices in MC-VQA (e.g., $\{A, B, C, D\}$) allows for a clean analysis while avoiding the noisy and potentially biased process of manually clustering open-ended answers, or training auxiliary projection heads to map LVLM representations into pre-defined clusters. Such additional components make PID estimates sensitive to clustering and projection hyperparameters, making it unclear whether the estimated quantities primarily reflect the LVLM's original end-to-end behavior or the added mapping, which is not how these models are typically used.

As illustrated in Figure 1 (1), our pipeline begins by defining the source variables—vision ($X_1$) and language ($X_2$)—from an LVLM's internal embeddings. We then conduct multimodal and unimodal inference runs to obtain three conditional probability distributions: $P(Y \mid X_1, X_2)$, $P(Y \mid X_1)$, and $P(Y \mid X_2)$. These distributions, along with the source features $X_1$ and $X_2$, are then fed into the BATCH estimator to compute the final PID values $\{R, U_1, U_2, S\}$. Notably, this estimation relies only on the model's predictive distributions and input representations; no ground-truth labels are used when computing the PID components. This makes PID a process-level descriptor of model behavior, complementary to standard accuracy metrics.

**Input representation and unimodal conditioning.** We define the source variables $X_1$ and $X_2$ as the mean-pooled visual and textual token embeddings, respectively, and ablate alternative summarization strategies (last-hidden and max pooling) in Appendix B.

Estimating the unimodal conditionals $P(Y \mid X_1)$ and $P(Y \mid X_2)$ for an integrated LVLM requires a carefully designed probe. We approximate unimodal conditioning by masking one modality at the embedding level: following the corruption scheme of Meng et al. (2022), we replace the entire embedding sequence of the other modality with noise. Each vector in this noise sequence is drawn i.i.d. from $\mathcal{N}(\boldsymbol{\mu}, \mathrm{diag}(\boldsymbol{\sigma}^2))$, where $\boldsymbol{\mu}, \boldsymbol{\sigma} \in \mathbb{R}^d$ denote the dimension-wise mean and standard deviation of that modality's embeddings, pre-computed across the dataset. This calibrated noise removes the other modality while keeping the embedding scale in-distribution.

**Confidence thresholding for renormalization.** For both unimodal and multimodal conditioning, we extract a categorical predictive distribution over the finite candidate set $\mathcal{Y}$. We first compute a token-length normalized candidate score $S_{\mathrm{orig}}(Y{=}y \mid \cdot)$ from the model log-likelihood of the candidate answer string, and then renormalize across candidates to obtain $P(Y \mid \cdot)$.

However, renormalizing over a restricted candidate set can artificially inflate confidence when the model assigns low scores to all candidates under the full vocabulary distribution. To mitigate overconfidence from a restricted candidate set, we compute the total candidate-set score and apply a confidence threshold:

$$\hat{P}(Y \mid \cdot) = \begin{cases} P(Y \mid \cdot) & \text{if } \sum_{y \in \mathcal{Y}} S_{\mathrm{orig}}(Y{=}y \mid \cdot) \geq \tau \\ \mathcal{U}(K) & \text{otherwise} \end{cases} \tag{5}$$

where $K = |\mathcal{Y}|$ and $\mathcal{U}(K)$ denotes the uniform distribution over $\mathcal{Y}$. This prevents low-confidence guesses from contributing spurious structure to the PID computation. We also ablate the confidence threshold $\tau \in \{0.2, 0.3, 0.4\}$; see Appendix B for details.

**Soft aggregation for the marginal output distribution.** A final adaptation addresses the estimation of the marginal output distribution $P(Y)$. Discretizing predictions via argmax and then computing frequencies can introduce a measurement artifact: for a totally uncertain (uniform) output, argmax resolves ties in a fixed manner (e.g., always selecting the first label), artificially converting uncertainty into a sharp peak upon aggregation. To avoid this, we use soft aggregation and estimate $P(Y)$ by averaging the regularized predictive distributions across all $N$ samples:

$$P(Y) = \frac{1}{N} \sum_{i=1}^{N} \hat{P}_i(Y), \tag{6}$$

where $\hat{P}_i(Y) = \hat{P}(Y \mid \cdot)_i$ denotes the regularized categorical distribution for sample $i$. This preserves the model's output statistics and leads to a more faithful PID analysis.

## 3.3 ANALYSIS DIMENSIONS & EXPERIMENTAL SETTINGS

To conduct a comprehensive information-decomposition analysis, we design experiments across three dimensions: (1) a large-scale comparison across a wide range of models and tasks, (2) the layer-wise information flow inside representative models, and (3) the learning dynamics by examining model checkpoints throughout the training process. For reproducibility, all experimental settings, including inference details and key hyperparameters, are provided in Appendix C.

### 3.3.1 CROSS-MODEL AND CROSS-TASK COMPARISON

**Models.** To assess how architecture and scale affect information use, we analyze 26 models (0.5B to 90B parameters) from 11 open-source LVLM families[1]. Our selection prioritizes recent, state-of-the-art families including LLaVA-OneVision (Li et al., 2024), Qwen2-VL (Wang et al., 2024), Qwen2.5-VL (Bai et al., 2025a), Gemma-3 (Team et al., 2025), InternVL2.5 (Chen et al., 2024), InternVL3 (Zhu et al., 2025a), and Llama-3.2-Vision (Meta AI, 2024). We also include Cambrian-1 (Tong et al., 2024) for its multi-vision encoder design and established models like LLaVA-1.5 (Liu et al., 2023a), InstructBLIP (Dai et al., 2023), and Fuyu (Bavishi et al., 2023) to serve as baselines.

---

[1]All model checkpoints are downloaded from Hugging Face: `https://huggingface.co/models`.

**Tasks.** We evaluate all models on four diverse MC-VQA datasets: MMBench ('en_dev') (Liu et al., 2024) for general reasoning, POPE ('COCO14 adversarial') (Li et al., 2023b) and Reefknot ('Perception & MCQ') (Zheng et al., 2024) for hallucination evaluation, and PMC-VQA ('test_clean') (Zhang et al., 2023) for domain-specific medical knowledge. Table 1 summarizes the characteristics.

Table 1: Details of the datasets used for evaluation. The listed training and test splits are not for LVLM fine-tuning; they are created by randomly partitioning each dataset (3:1 ratio) for the PID estimation, as the BATCH estimator requires separate sets to train networks and estimate PID values.

| Dataset | Task | # of options | # of training samples | # of test samples |
|---------|------|--------------|-----------------------|-------------------|
| MMBench | General visual reasoning | 2–4 | 3246 | 1083 |
| POPE | Hallucination evaluation | 2 | 2250 | 750 |
| Reefknot | Hallucination evaluation | 4 | 1612 | 538 |
| PMC-VQA | Domain-specific knowledge | 4 | 1500 | 500 |

**Image-removal intervention.** As a behavioral validation, we remove the image to obtain a text-only baseline and measure the accuracy drop $D_{\text{vision}}$, which we relate to the PID-based information spectrum to assess models' visual reliance.

### 3.3.2 Layer-wise information dynamics

To trace the internal information flow, we conduct a layer-wise PID analysis on three representative model families: InternVL3-2B/8B, Qwen2.5-VL-3B/7B, and LLaVA-1.5-7B/13B. We analyze them on MMBench and PMC-VQA to observe dynamics on general and domain-specific tasks. By applying the *logit lens* (nostalgebraist, 2020), we project the hidden state at each transformer block through the LM head to obtain a layer-specific output distribution for our PID analysis.

### 3.3.3 Learning dynamics of multimodal fusion

To understand how fusion capabilities evolve, we analyze the two-stage training process of a representative model, LLaVA-1.5 (7B/13B), and reproduce its training using the original data and official settings. This process involves (1) vision-language alignment pretraining, where only the projector is trained to align frozen vision and language embeddings, followed by (2) visual instruction tuning, which fine-tunes both the projector and the LLM.

We save four equidistant checkpoints from each stage and evaluate each checkpoint's full PID profile on MMBench and PMC-VQA to create a temporal trace of its learning trajectory.

## 4 Results and findings

We treat PID components as signals: $R$ (overlap), $U_1$ (vision-only cues), $U_2$ (language-side knowledge), and $S$ (combined use). Across **breadth (models $\times$ datasets)**, **depth (layers)**, and **time (training)**, we ask which signal most consistently shapes LVLM behavior and generalization.

### 4.1 Dimension 1: Cross-model & cross-task comparison

Because redundancy $R$ and vision uniqueness $U_1$ are consistently small, we focus on synergy $S$ and language uniqueness $U_2$ to characterize task demand and model strategy (see full spectra in Appendix E.1).

### 4.1.1 Two regimes of information use across tasks

**Task-level patterns: two regimes of evidence use.** *Question: Do datasets push LVLMs to rely more on combining inputs, or on what they already know from text?* To understand how different tasks challenge LVLMs, we first investigate whether they systematically demand different kinds of

information. Our analysis examines how models allocate information between $S$ and $U_2$ on each dataset, revealing recurring differences that we summarize as two regimes of information use.

Figure 2 plots, for each dataset, the dataset-level mean shares of $S$ and $U_2$ averaged across all models, with 95% bootstrap CIs. A clear split emerges, driven mainly by $S$: MMBench and POPE form one cluster characterized by high $S$, while Reefknot and PMC-VQA form a second cluster with markedly lower $S$ and higher $U_2$. These are empirical profiles of how LVLMs behave on these datasets, not labels of the datasets themselves. In this second cluster, synergy appears to have a practical ceiling, suggesting that while fusion is beneficial, it cannot fully compensate for missing language-side knowledge; correspondingly, accuracies are typically 20–30% lower than in the high-$S$ group.

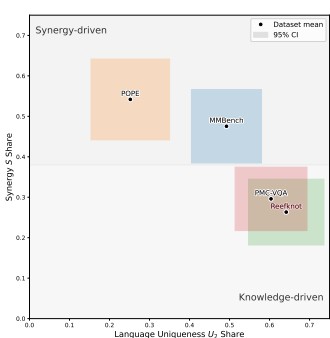

Figure 2: Share of synergy $S$ and language uniqueness $U_2$ across four datasets.

> **Finding 1:** Across the four benchmarks, we observe two recurring information-use regimes in our experiments. A **synergy-driven regime** appears on tasks where models rely more on strong cross-modal fusion. In contrast, a **knowledge-driven regime** appears on tasks where performance reflects a stronger dependence on language-side knowledge and priors.

**Information correlates of accuracy across task regimes.** *Within each regime, we test which component tracks model accuracy.* We compute Spearman's $\rho$ between accuracy and each PID term across 26 LVLMs per dataset. Note that PID is computed from model predictions only; labels are used solely to report accuracy.

Table 2: Spearman correlations ($\rho$) and $p$-values across datasets.

| **Dataset** | $S$ | | $U_2$ | | $I(X_1, X_2; Y)$ | | $I(X_1; X_2; Y)$ | |
|---|---|---|---|---|---|---|---|---|
| | $\rho$ | $p$-**val** | $\rho$ | $p$-**val** | $\rho$ | $p$-**val** | $\rho$ | $p$-**val** |
| MMBench | **0.750** | $< 0.001$ | 0.194 | 0.343 | **0.632** | $< 0.001$ | **-0.757** | $< 0.001$ |
| POPE | 0.742 | $< 0.001$ | -0.009 | 0.964 | 0.157 | 0.445 | -0.701 | $< 0.001$ |
| Reefknot | 0.357 | 0.073 | 0.313 | 0.119 | 0.266 | 0.196 | -0.348 | 0.081 |
| PMC-VQA | 0.432 | 0.027 | **0.406** | 0.040 | 0.559 | 0.003 | -0.587 | 0.002 |

Table 2 shows a consistent pattern. On synergy-driven benchmarks (MMBench, POPE), $S$ is the strongest positive correlate of accuracy ($\rho \approx 0.75$, $p < 0.001$), whereas $I(X_1, X_2; Y)$ is less consistent across datasets.[2] This implies that top-performing models are not those with simply "more" information, but those that translate overlapping cues into effective cross-modal use.

On knowledge-driven benchmarks (Reefknot, PMC-VQA), the picture shifts: $U_2$ becomes comparatively more informative (significant on PMC-VQA), while $S$ remains positively related to accuracy but is no longer dominant. These results suggest that fusion is beneficial in both regimes, but gains are bounded when language-side knowledge becomes the primary bottleneck.

> **Finding 2:** In synergy-driven tasks, accuracy is most strongly associated with synergy $S$. In knowledge-driven tasks, language uniqueness $U_2$ becomes more predictive, while synergy $S$ remains helpful but less dominant.

---

[2]The interaction information $I(X_1; X_2; Y) = R - S$ is strongly negative here, largely mirroring the $-S$ term.

**Intervention-based validation: image removal.** We further validate this interpretation via a simple intervention: removing the image and measuring the accuracy drop $D_{\text{vision}}$. Across models, $D_{\text{vision}}$ correlates strongly with $S$ on synergy-driven benchmarks (MMBench/POPE: $\rho = 0.809/0.744$, both with $p < 0.001$), and more weakly on knowledge-driven ones (Reefknot/PMC-VQA: $\rho = 0.459/0.400$, $p = 0.018/0.043$). This confirms a key prediction: models with higher $S$ are more sensitive to visual ablation, indicating that $S$ captures decision-relevant visual reliance.

Qualitative examples illustrating $S$-dominant (MMBench/POPE) and $U_2$-bounded (Reefknot/PMC-VQA) cases are provided in Appendix D. We next ask whether such accuracy-relevant components reflect consistent model-level strategies.

### 4.1.2 INFORMATION STRATEGIES ACROSS MODEL ARCHITECTURES

**Model families exhibit stable, contrasting information strategies.** *Do model families lean toward combining inputs or toward language knowledge—and does that preference hold across settings?* We summarize each family's behavior by its median $S$ and $U_2$ within each regime, shown in Figure 3.

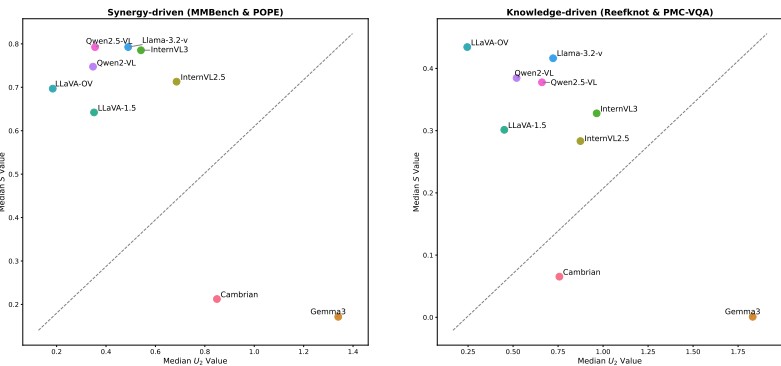

Figure 3: Family-level strategies: median $S$ versus median $U_2$ per family, computed across model sizes within each task regime. Points show the family medians for each regime. Outliers (Instruct-BLIP, Fuyu) are omitted for clarity.

Families occupy two clearly separated regions, corresponding to two information-use strategies: a **fusion-centric** group (e.g., InternVL2.5/3, Qwen2/2.5-VL) with relatively high $S$ and lower $U_2$, and a **language-centric** group (e.g., Gemma3, Cambrian) with lower $S$ and higher $U_2$. Although absolute $S$ drops on knowledge-driven tasks, the relative positions of families remain similar across regimes, suggesting that this preference is a stable family-level tendency.

> **Finding 3:** LVLM families exhibit two stable information strategies: **fusion-centric** families rely more on synergy $S$, while **language-centric** families rely more on language uniqueness $U_2$. This strategic identity persists across task regimes.

**Scaling effects on synergy-driven tasks.** *If family identity is stable, scaling should reinforce the same leaning. A common expectation is that larger models rely more on $U_2$; we test this on synergy-driven tasks.* To assess how information use changes with scale, we compare Small (S), Mid (M), and Very-Large (VL) models within representative families.

As shown in Table 3, the share of language uniqueness $U_2$ does not systematically increase with size and often decreases. By contrast, the accuracy differences between sizes tend to co-vary with changes in the share of synergy $S$: larger checkpoints that improve more in accuracy also exhibit larger increases in $S$. This is consistent with Finding 2, where on synergy-driven tasks performance is more closely tied to $S$ than to $U_2$.

Table 3: Scaling on synergy-driven tasks: changes in accuracy (ΔAcc) and PID shares ($\Delta S$, $\Delta U_2$) for S→M and M→VL within representative families.

| Family | Sizes (B) | S→M (%) | | | M→VL (%) | | |
|---|---|---|---|---|---|---|---|
| | | ΔAcc | $\Delta S$ | $\Delta U_2$ | ΔAcc | $\Delta S$ | $\Delta U_2$ |
| LLaVA-OneVision | 0.5→7→72 | 11.9 | 11.9 | -6.5 | 3.1 | 14.8 | -9.7 |
| Qwen2-VL | 2→7→72 | 5.7 | 0.6 | 8.4 | -3.9 | 0.4 | -0.6 |
| Qwen2.5-VL | 3→7→72 | 1.3 | -6.3 | 1.0 | 2.5 | 5.5 | 2.5 |
| InternVL2.5 | 2→8→78 | 7.3 | 36.8 | -55.6 | 3.6 | 10.6 | 3.8 |
| InternVL3 | 2→8→78 | 2.7 | 2.5 | -6.2 | 6.4 | 4.6 | -10.3 |

> **Finding 4:** On synergy-driven tasks, within each family the accuracy differences between available sizes are more closely associated with changes in $S$ than with changes in $U_2$. Larger checkpoints therefore tend to benefit more from stronger multimodal fusion than from further amplifying language-side priors.

## 4.2 DIMENSION 2: LAYER-WISE INFORMATION DYNAMICS

*Because stable family preferences and gains with scale in Dimension 1 tracked increases in S, the next question is where S arises in the stack.* We therefore analyze layer-wise information with the *logit lens*; full results are in Appendix E.2.

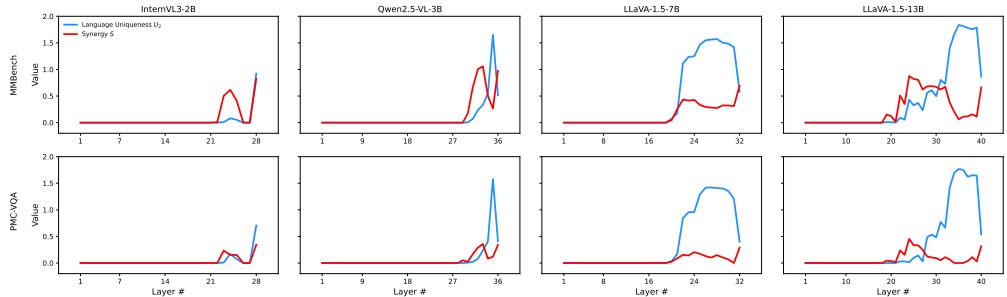

Figure 4: Layer-wise PID dynamics for representative models on synergy-driven (MMBench, top) and knowledge-driven (PMC-VQA, bottom) tasks. A consistent three-phase pattern appears across models and datasets.

For Qwen2.5-VL-7B and InternVL3-8B, since the logit lens does not yield meaningful intermediate predictions, we omit them here. Figure 4 shows a consistent three-phase profile across models and datasets. $S$ typically emerges and peaks in the middle-to-late layers, often softens near the output, then spikes at the final layer. $U_2$ generally builds through the stack, peaking at the second-to-last layer before a sharp final drop. $R$ and $U_1$ remain small throughout. An exception is InternVL3-2B, where $U_2$ does not exhibit the final drop.

> **Finding 5:** The layer-wise dynamics reveal a shared three-phase pattern of information flow. Information emerges in the middle-to-late layers, then moves from language-based representation building in the later layers to a decisive, synergistic fusion of modalities in the final layer.

## 4.3 DIMENSION 3: LEARNING DYNAMICS OF MULTIMODAL FUSION

*While layer-wise snapshots reveal **where** S arises, they do not show **when** it emerges. We therefore turn to the training trajectory.* We trace PID through the two-stage training of LLaVA-1.5 (7B, 13B). Full results are in Appendix E.3.

The results in Figure 5 show a clear separation between the two training stages in our reproduced LLaVA-1.5 pipeline. Throughout alignment pretraining (Stage 1), both $S$ and $U_2$ remain low and relatively stable. Once visual instruction tuning begins (Stage 2), both components increase markedly. The effect of model scale also differs across components: the 7B model shows a more pronounced increase in $S$, whereas the 13B model exhibits a stronger increase in $U_2$, indicating that larger models in this setting place greater emphasis on language-side priors during fine-tuning. Overall, both trends suggest that fine-tuning benefits from scale.

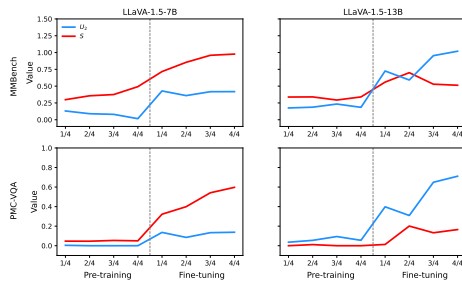

Figure 5: Evolution of $S$ and $U_2$ during two-stage training of LLaVA-1.5 (7B, 13B).

This aligns with observations from prior work such as MiniGPT-4 (Zhu et al., 2023), where the second-stage visual instruction tuning was found to be crucial for improving generation quality. Our PID analysis provides an information-theoretic view of this phenomenon, showing how fusion $S$ and language priors $U_2$ emerge and diverge across stages and model scales.

> **Finding 6:** In the two-stage training of LLaVA-1.5, multimodal fusion $S$ remains negligible during alignment pretraining and increases primarily during visual instruction tuning. Across the two available sizes, the 7B model shows relatively larger gains in $S$, whereas the 13B model exhibits comparatively stronger growth in $U_2$ during this stage.

**Summary of empirical findings.** Taken together, our results show that PID provides a coherent view of LVLM behavior across three complementary axes. At the task level, benchmarks fall into recurring information-use regimes, and at the family level, model families adopt contrasting strategies characterized by different balances of $S$ and $U_2$; at the layer level, $S$ and $U_2$ follow a shared three-phase pattern of information flow; and across training stages, multimodal fusion $S$ emerges primarily during visual instruction tuning and interacts with model scale.

## 5 CONCLUSION

LVLMs are typically evaluated by accuracy, which tells us *what* they get right but not *how* different modalities are utilized. In this work, we introduce a PID-based framework that yields a process-level decomposition of decision-relevant information in LVLMs and, to our knowledge, provides the first systematic application of PID at this scale. By adapting a scalable PID estimator to LVLM outputs and applying it to 26 models across four benchmarks, we offer an information-theoretic lens that complements accuracy-only evaluation and supports more targeted analysis of multimodal behavior.

This study has several **limitations**. (1) PID estimation assumes a discrete target space, so we do not cover fully open-ended generation tasks. (2) Our unimodal probes are approximate: masking a modality with calibrated noise stabilizes estimation, but $U_1$, $U_2$, and $S$ are measured under this probe rather than under truly natural unimodal inputs. (3) PID is correlational: the components are derived from model predictions and inputs, and their relationships to accuracy or interventions reflect associations rather than full causal mechanisms.

**Future work** can extend this study in several directions:

1. **Methodology:** developing PID estimators and output encodings that handle richer generative settings and additional modalities, and exploring complementary unimodal probes.
2. **Model and training design:** using $(U_1, U_2, S)$ as diagnostic signals during scaling and instruction tuning, and potentially as auxiliary objectives to balance fusion and language priors.
3. **Evaluation:** using PID-based analyses to guide the construction of benchmarks that explicitly require high synergy $S$ or isolate language priors $U_2$.

ACKNOWLEDGMENTS

This work was supported by JST-SPRING Grant Number JPMJSP2108, JST-CRONOS Grant Number JPMJCS24K8, JSPS KAKENHI Grant Number JP23K28139, and the Institute of AI and Beyond of the University of Tokyo.

ETHICS STATEMENT

All authors have read and follow the ICLR Code of Ethics. Our study analyzes publicly available LVLM checkpoints and datasets (MMBench, POPE, Reefknot and PMC-VQA); no new data were collected, no human subjects were involved, and no personally identifiable information is used. We respect dataset/model licenses and cite original sources; we do not redistribute proprietary content. The work focuses on interpretability/analysis (PID of information use) rather than deployment, and is intended to improve transparency of multimodal systems. Potential risks (e.g., inherited dataset or model biases) are acknowledged; we report cases in the paper/appendix. The authors declare no conflicts of interest and no sponsorship that influenced the results.

REPRODUCIBILITY STATEMENT

Our proposed PID framework and experimental design are described in Section 3. Full tables/plots and additional analyses appear in Appendix E and robustness checks in Appendix B. The public repository linked in the abstract contains code and configuration files needed to rerun the study, including dataset splits, prompts, preprocessing steps, model versions, and random seeds, as well as scripts to compute PID.

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

LLM USAGE

We used an LLM-based assistant only for writing support, including sentence paraphrasing, grammar/typo checking, and outline/flow refinement. The LLM was not used to design experiments, analyze data, or generate results; all technical content (methods, equations, figures, and code) was produced and verified by the authors. LLM-suggested text was reviewed and edited for accuracy, and references were inserted and checked manually.

## A  PARTIAL INFORMATION DECOMPOSITION AND ITS ESTIMATION

### A.1  INTRODUCTION OF PARTIAL INFORMATION DECOMPOSITION

Classical mutual information (MI) quantifies the information a single variable provides about another: $I(X;Y)$ measures the reduction in uncertainty from $H(Y)$ to $H(Y \mid X)$ (Shannon, 1948). Extending this analysis to a system with multiple source variables—for instance, two sources $X_1$ and $X_2$ with state spaces $\mathcal{X}_1$ and $\mathcal{X}_2$, and a target $Y$ with state space $\mathcal{Y}$—is challenging, as the standard interaction information, $I(X_1; X_2; Y)$, can be positive or negative. This sign ambiguity complicates its interpretation and motivates a decomposition of information into a set of well-behaved, non-negative quantities.

Partial information decomposition (PID), first proposed by Williams & Beer (2010), addresses this issue. We adopt the definition from Bertschinger et al. (2014), which decomposes the total information into three conceptual components: redundancy, uniqueness, and synergy. These concepts are quantified by four non-negative atoms: redundant information $(R)$, unique information from the first source $(U_1)$, unique information from the second source $(U_2)$, and synergistic information $(S)$.

PID postulates the following consistency relations linking the four atoms to 4 classical mutual information terms:

$$I(X_1, X_2; Y) = R + U_1 + U_2 + S, \tag{7}$$
$$I(X_1; Y) = R + U_1, \tag{8}$$
$$I(X_2; Y) = R + U_2, \tag{9}$$
$$I(X_1; X_2; Y) := I(X_1; Y) + I(X_2; Y) - I(X_1, X_2; Y) = R - S. \tag{10}$$

Eqs. 7–10 establish the algebraic relationship between $(R, U_1, U_2, S)$ and the usual information measures, cleanly separating redundancy $(R)$ from synergy $(S)$ through the co-information identity Eq. 10.

The definition for the PID atoms relies on optimization over the set of *marginal-matching* distributions:

$$\Delta_P = \big\{ Q \in \Delta : Q(x_i, y) = P(x_i, y), \ \forall\, x_i \in \mathcal{X}_i, \ y \in \mathcal{Y}, \ i \in \{1, 2\} \big\}, \tag{11}$$

where $\Delta$ is the space of all possible joint distributions of $X_1$, $X_2$ and $Y$. This set contains all distributions $Q$ that preserve the marginal information of the true distribution $P$, while allowing other dependencies to vary. Let $I_Q(\cdot)$ denote mutual information under a distribution $Q$, the PID atoms are defined as:

$$R = \max_{Q \in \Delta_P} I_Q(X_1; X_2; Y), \tag{12}$$

$$U_1 = \min_{Q \in \Delta_P} I_Q(X_1; Y \mid X_2), \tag{13}$$

$$U_2 = \min_{Q \in \Delta_P} I_Q(X_2; Y \mid X_1), \tag{14}$$

$$S = I(X_1, X_2; Y) - \min_{Q \in \Delta_P} I_Q(X_1, X_2; Y). \tag{15}$$

Intuitively, optimizing over $\Delta_P$ isolates each component of information. For instance, a distribution $Q^\star \in \Delta_P$ that minimizes the total mutual information $I_Q(X_1, X_2; Y)$ does so by reducing the higher-order (synergistic/complementary) dependencies while preserving the source–target marginals. The resulting gap, used to define synergy in Eq. 15, therefore measures this emergent

information. Similarly, the unique information components (Eqs. 13 and 14) represent the minimal necessary conditional information from each source, while redundancy (Eq. 12) represents the maximal possible shared co-information.

Under this construction, the four atoms are non-negative and satisfy the axiomatic relations in Eqs. 7–10. A key insight from Bertschinger et al. (2014) is that the optimization problems defining $R, U_1, U_2$, and $S$ are *equivalent*, and it is sufficient to solve one of them.

### A.2 THE ESTIMATOR WE LEVERAGE: BATCH FOR CONTINUOUS REPRESENTATIONS

We adopt BATCH, a scalable PID estimator for high-dimensional, continuous input representations $X_1$ and $X_2$ and large datasets (Liang et al., 2023a). The method amortizes the optimization problems in Eqs. 12–15 by learning a parametric joint $\tilde{Q}(x_1, x_2, y)$ that lies in the marginal-matching set $\Delta_P$ defined earlier. Specifically, $\tilde{Q}$ is trained to approximately solve

$$\min_{Q \in \Delta_P} I_Q(X_1, X_2; Y), \tag{16}$$

and, by the equivalence of the optimization characterizations, the remaining PID components are obtained by evaluating the corresponding quantities under the same $\tilde{Q}$.

**Neural parameterization and projection.** Given mini-batches $(\mathbf{X}_1, \mathbf{X}_2, \mathbf{Y})$, two encoders $f_{\phi(1)}$ and $f_{\phi(2)}$ output features that define an *unnormalized* joint via a similarity matrix:

$$A = \exp\big(f_{\phi(1)}(\mathbf{X}_1, y) \, f_{\phi(2)}(\mathbf{X}_2, y)^\top\big), \tag{17}$$

where $A[i][j][y] = \tilde{Q}(\mathbf{X}_1[i], \mathbf{X}_2[j], y)$ for each $y \in \mathcal{Y}$. To enforce the marginal constraints $Q \in \Delta_P$, BATCH applies the Sinkhorn-Knopp algorithm (Cuturi, 2013) to iteratively normalize rows and columns of $A$ so the projected distribution matches the fixed pairwise marginals $P(x_1, y)$ and $P(x_2, y)$.

**Training objective.** Given matrix $A$ representing $\tilde{Q}(x_1, x_2, y)$, the objective can be written as:

$$\min_{Q \in \Delta_P} \mathbb{E}_{\substack{x_1, y \sim Q(x_1, y) \\ x_2 \sim Q(x_2|x_1, y)}} \left[ \log \frac{Q(x_2 \mid x_1, y) \, Q(x_1 \mid y)}{\sum_{y' \in Y} Q(x_2 \mid x_1, y') \, Q(y' \mid x_1) \, Q(x_1)} \right], \tag{18}$$

hence gradient descent is leveraged to optimize this via updating the parameters of $f_{\phi(1)}$ and $f_{\phi(2)}$.

**Estimating PID values via learned models.** Upon convergence, we estimate the required information terms under the data distribution $P$ and the learned $\tilde{Q}$. Using the consistency relations and the optimization equivalence, the PID components are obtained as

$$R = I_{\tilde{Q}}(Y; X_1, X_2), \tag{19}$$

$$U_1 = I_{\tilde{Q}}(Y; X_1, X_2) - I_P(Y; X_2), \tag{20}$$

$$U_2 = I_{\tilde{Q}}(Y; X_1, X_2) - I_P(Y; X_1), \tag{21}$$

$$S = I_P(Y; X_1, X_2) - I_{\tilde{Q}}(Y; X_1, X_2). \tag{22}$$

These quantities satisfy the PID consistency equations by construction and recover the optimization-defined components when $\tilde{Q}$ is learned.

**Rationale for adopting the BATCH estimator.** The BATCH estimator provides a practical and scalable method for calculating these PID atoms from data. The core of this approach is to use neural networks to parameterize and learn an approximate joint distribution, $\tilde{Q}$, that satisfies the required marginal-matching constraints. By optimizing an information-theoretic objective over mini-batches, the estimator can be effectively applied to large datasets. We chose to adapt this estimator for two primary reasons. First, it was explicitly designed for general multimodal learning contexts. Second, and most importantly, it operates on high-dimensional, continuous features. This latter property makes it uniquely suited for analyzing modern LVLMs, as our framework can apply it directly to the rich vector embeddings these models produce to quantify their internal information dynamics.

# B    ABLATION STUDY

To validate our methodology, we examine sensitivity to two implementation choices: (i) feature summarization (mean pooling, last-hidden state, and max pooling) and (ii) the confidence threshold $\tau \in \{0.2, 0.3, 0.4\}$. We evaluate four representative LVLMs chosen to span *families*, *scales*, and *strategy types*: Qwen2.5-VL-7B and Qwen2.5-VL-72B (fusion-centric, two scales of the same family), Gemma3-4B (language-centric), and Cambrian-34B (language-centric with more parameters). Ablations are run on the synergy-driven MMBench and the knowledge-driven PMC-VQA datasets. Because $S$ and $U_2$ are the primary components in these regimes, we report two summary tables: synergy $S$ on MMBench (Table 4) and language uniqueness $U_2$ on PMC-VQA (Table 5).

Table 4: $S$ on MMBench for four chosen models under two ablations (feature summarization and confidence threshold).

| Model | Feature summarization | | | Confidence threshold $\tau$ | | |
|---|---|---|---|---|---|---|
| | Mean (ours) | Last-hidden | Max-pool | 0.3 (ours) | 0.2 | 0.4 |
| Qwen2.5-VL-7B | 1.112 | 1.112 | 1.112 | 1.112 | 1.112 | 1.112 |
| Qwen2.5-VL-72B | 1.088 | 1.088 | 1.088 | 1.088 | 1.088 | 1.088 |
| Gemma3-4B | 0.167 | 0.172 | 0.173 | 0.167 | 0.167 | 0.167 |
| Cambrian-34B | 0.630 | 0.637 | 0.630 | 0.630 | 0.606 | 0.630 |

Table 5: $U_2$ on PMC-VQA for four chosen models under two ablations (feature summarization and confidence threshold).

| Model | Feature summarization | | | Confidence threshold $\tau$ | | |
|---|---|---|---|---|---|---|
| | Mean (ours) | Last-hidden | Max-pool | 0.3 (ours) | 0.2 | 0.4 |
| Qwen2.5-VL-7B | 0.665 | 0.665 | 0.665 | 0.665 | 0.665 | 0.665 |
| Qwen2.5-VL-72B | 0.893 | 0.893 | 0.893 | 0.893 | 0.893 | 0.893 |
| Gemma3-4B | 1.864 | 1.864 | 1.864 | 1.864 | 1.864 | 1.864 |
| Cambrian-34B | 0.698 | 0.698 | 0.698 | 0.698 | 0.698 | 0.698 |

**Input feature summarization.** We compare mean pooling (used in the main experiments) with two common alternatives: **last-hidden state** and **max pooling**. On MMBench (Table 4), $S$ is unchanged for the two Qwen2.5-VL models; Gemma3-4B varies slightly ($0.167 \to 0.173$), and Cambrian-34B varies within 0.630–0.637. On PMC-VQA (Table 5), $U_2$ is identical across all pooling choices for all models. Thus, feature summarization has negligible effect on the components most relevant to each regime.

**Confidence threshold $\tau$.** We vary the regularization threshold around our default ($\tau = 0.3$) to $\tau \in \{0.2, 0.4\}$. On MMBench (Table 4), $S$ is invariant for both Qwen2.5-VL models and for Gemma3-4B; Cambrian-34B shows a small dip at $\tau = 0.2$ (0.606 vs. 0.630 at $\tau = 0.3/0.4$). On PMC-VQA (Table 5), $U_2$ is unchanged across $\tau$ for all models. These results indicate stability of our conclusions with respect to the confidence-regularization setting in the tested range.

**Summary.** Across both ablations, the regime-defining components ($S$ on MMBench, $U_2$ on PMC-VQA) remain effectively constant, supporting the robustness of our methodology.

## C  DETAILED EXPERIMENTAL SETTINGS

All experiments reported in this paper, including model inference, PID estimation, and training reproduction, were conducted on servers equipped with 8 NVIDIA A100 GPUs.

**General inference details.**  For all multiple-choice VQA tasks, we use a standardized prompt that instructs the model to answer with only the letter of the correct option: "Please select the correct answer from the options above. You must answer with the letter of the correct option only." While most modern LVLMs adhere to this instruction, some earlier models (e.g., InstructBLIP, Fuyu-8b) tend to generate conversational, free-form text.

To handle these inconsistencies, our reported "accuracy" is not a standard logit-based metric. Instead, we perform a strict string match on the first generated token: a prediction is correct only if the normalized token (lowercased, punctuation removed) exactly matches the ground-truth letter. This format-dependent evaluation means the performance floor is 0%, not the random-guess rate, as models that fail to follow the required format will be marked incorrect. For reproducible outputs, we use a deterministic greedy decoding strategy (no sampling) for all models.

**Hyperparameters for BATCH estimator.**  Although the BATCH estimator is known to be robust to hyperparameter settings (Liang et al., 2023a), we adhere to the original configuration for consistency and reproducibility. The key hyperparameters used for the estimator's neural networks are listed in Table 6.

Table 6: Hyperparameters for the BATCH Estimator.

| Hyperparameter | Value |
|---|---|
| Learning rate | 1e-3 |
| Optimizer | Adam |
| Number of epochs | 8 |
| Network architecture | 3-layer MLP |
| Hidden dimension | 32 |
| Activation function | ReLU |
| Training batch size | 256 |
| Test batch size | 256 |

**Hyperparameters for reproducing LLaVA-1.5 training.**  We exactly follow the official two-stage recipe of Liu et al. (2023a) with **no changes** to data, model, or optimization hyperparameters. For analysis, we saved four equally spaced checkpoints from each stage and evaluated with greedy decoding (sampling disabled). For full hyperparameters, see Table 9 of Liu et al. (2023a).

# D    CASE STUDIES

To provide a more qualitative understanding of our findings, we visualize the PID results for two representative examples. For each VQA pair, we show the outputs from four LVLMs that exemplify different strategies (fusion-centric vs. language-centric) and scales. These cases provide concrete illustrations of how different models use information to solve tasks from the two distinct regimes we identified.

**Case 1: A synergy-driven task (MMBench).**    This task requires the model to identify the state of Massachusetts on a map. Success depends on correctly associating the visual shape of the state with its name in the text—a classic fusion task where neither modality alone is sufficient.

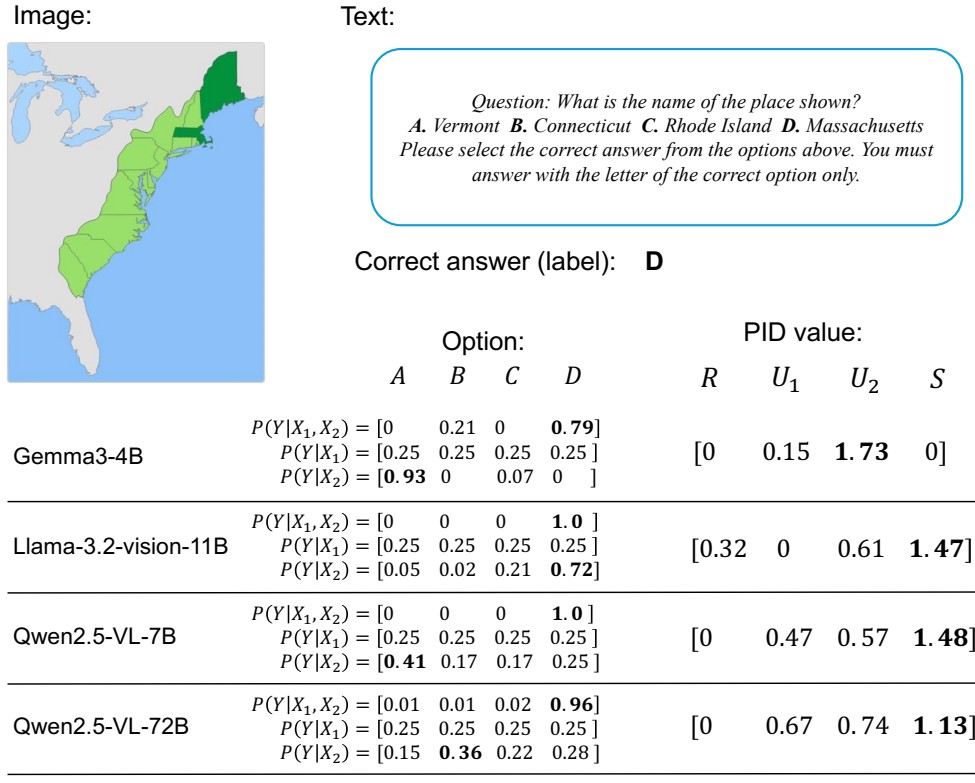

Figure 6: PID analysis for a synergy-driven task. All models answer correctly, but the PID results reveal two distinct solution strategies: generating high synergy (Llama-3.2-vision, Qwen2.5-VL) versus correcting a strong, incorrect language prior (Gemma3).

As shown in Figure 6, all models arrive at the correct answer, but their methods differ dramatically. The fusion-centric models (Llama-3.2-vision and the Qwen2.5-VL series) solve the problem by generating a large amount of synergy $S$, confirming that the answer emerges from the direct interaction of image and text. In stark contrast, the language-centric Gemma model also answers correctly, but does so by overcoming a strong, incorrect language prior (a high $U_2$ favoring "Vermont"). Interestingly, the PID framework reveals this correction happens without generating synergy, showcasing a different path to success: the visual information acts to override a mistaken language bias, rather than creating new information with it. This highlights the framework's ability to distinguish between models that truly *fuse* modalities to create new insight and those that *arbitrate* between conflicting unimodal beliefs. This latter "correction" mechanism, where visual evidence $U_1$ overrides a strong language bias $U_2$, may represent an efficient, non-synergistic strategy common in smaller or more language-centric architectures.

**Case 2: A knowledge-driven task (Reefknot).** This task asks about the spatial relationship "through", which requires a nuanced understanding of prepositions that primarily resides within the language model. The visual context is relatively simple, but the linguistic concept is complex.

Figure 7 illustrates how these different strategies adapt to a knowledge-driven task, presenting a clear contrast to Case 1. As expected, the language-centric Gemma model solves the problem almost exclusively with its language prior (high $U_2$), generating no synergy. The fusion-centric models (Llama-3.2-vision and the Qwen2.5-VL family), however, tell a more interesting story: while they also depend heavily on language uniqueness, they continue to generate significant synergy (e.g., $S = 0.95$ for Qwen2.5-VL-7B).

This demonstrates a persistent strategic difference between model families. Even when a task seems solvable with language alone, fusion-centric models consistently attempt to integrate visual information to ground their linguistic understanding, whereas language-centric models default to their strong language priors.

Image:                Text:

Question: *What is the relation with car and window in this photo?*
**A.** *onto* **B.** *through* **C.** *on left of* **D.** *containing*
*Please select the correct answer from the options above. You must answer with the letter of the correct option only.*

Correct answer (label): **B**

|  | Option: | | | | PID value: | | | |
|---|---|---|---|---|---|---|---|---|
|  | $A$ | $B$ | $C$ | $D$ | $R$ | $U_1$ | $U_2$ | $S$ |
| Gemma3-4B | $P(Y\|X_1,X_2) = [0.24$ | **0.67** | $0.09$ | $0$ ] | $[0$ | $0.57$ | **1.87** | $0]$ |
|  | $P(Y\|X_1) = [0.25$ $0.25$ $0.25$ $0.25]$ | | | | | | | |
|  | $P(Y\|X_2) = [$**0.79** $0.18$ $0.03$ $0.]$ | | | | | | | |
| Llama-3.2-vision-11B | $P(Y\|X_1,X_2) = [0.03$ **0.8** $0.16$ $0.01]$ | | | | $[0.21$ | $0$ | **0.74** | $0.33]$ |
|  | $P(Y\|X_1) = [0.25$ $0.25$ $0.25$ $0.25]$ | | | | | | | |
|  | $P(Y\|X_2) = [0.01$ **0.77** $0.1$ $0.12]$ | | | | | | | |
| Qwen2.5-VL-7B | $P(Y\|X_1,X_2) = [0.01$ **0.95** $0.03$ $0.01]$ | | | | $[0$ | $0.60$ | $0.68$ | **0.95**] |
|  | $P(Y\|X_1) = [0.25$ $0.25$ $0.25$ $0.25]$ | | | | | | | |
|  | $P(Y\|X_2) = [0.18$ **0.38** $0.18$ $0.26]$ | | | | | | | |
| Qwen2.5-VL-72B | $P(Y\|X_1,X_2) = [0.02$ **0.91** $0.07$ $0$ ] | | | | $[0$ | $0.07$ | **1.11** | $0.41]$ |
|  | $P(Y\|X_1) = [0.25$ $0.25$ $0.25$ $0.25]$ | | | | | | | |
|  | $P(Y\|X_2) = [0.01$ $0.07$ $0.17$ **0.75**] | | | | | | | |

Figure 7: PID analysis for a knowledge-driven task. While all models rely heavily on language uniqueness $U_2$, fusion-centric models like Llama-3.2-vision and Qwen2.5-VL also generate non-trivial synergy, unlike the language-centric Gemma.

These case studies provide qualitative evidence for our quantitative findings, visually demonstrating how models with different core strategies can reach the same correct answer via entirely different information-processing pathways.

# E    FULL RESULTS

## E.1    FULL INFORMATION SPECTRA ON FOUR DATASETS

This section provides the full information spectra, including all four PID components $(R, U_1, U_2, S)$ and the corresponding accuracy for all 26 models across the four evaluated datasets. These figures supplement our main analysis in Section 4, where we focused on the two most discriminative components, synergy $S$ and language uniqueness $U_2$.

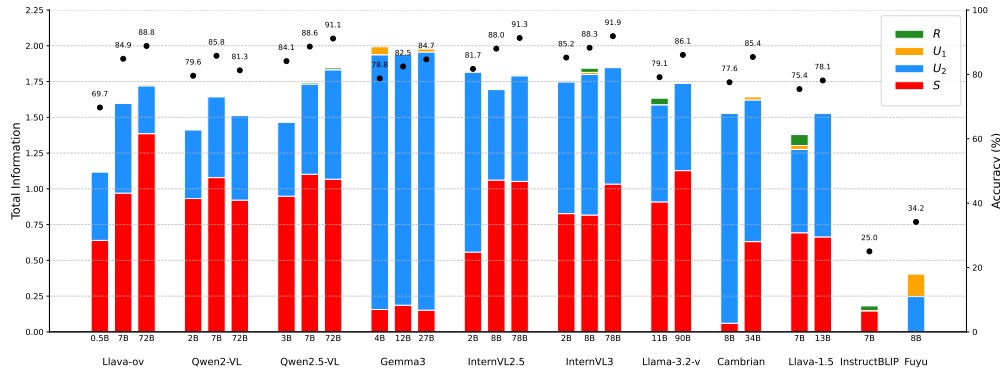

Figure 8: Full information spectrum and accuracy on the MMBench dataset. This figure provides detailed evidence for the synergy-driven regime. It visually confirms the existence of two primary model strategies: fusion-centric families (e.g., LLaVA-ov, Qwen2.5-VL, InternVL3) show a large share of synergy $S$, while language-centric families (e.g., Gemma3) are dominated by language uniqueness $U_2$. Furthermore, the plot illustrates that scaling models tends to increase synergy $S$ within many families.

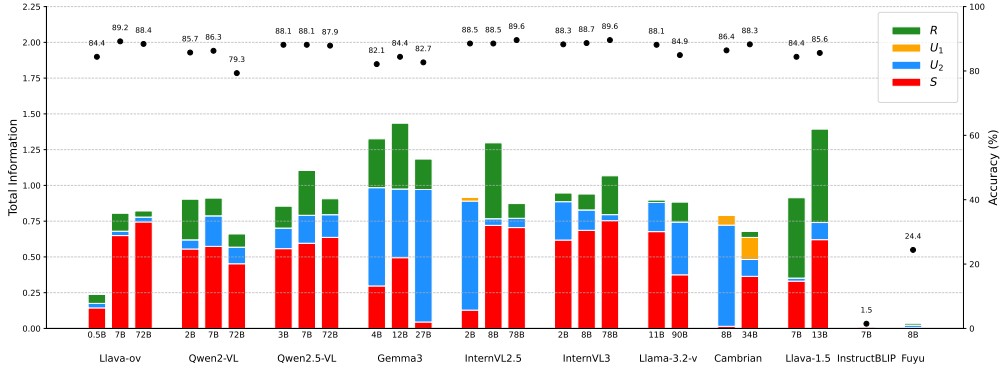

Figure 9: Full information spectrum and accuracy on the POPE dataset. As a synergy-driven task, POPE shows that synergy $S$ is a key component for many high-performing models. Uniquely, this dataset also elicits significant redundancy $R$ (green bars). This is likely because the task uses simple binary questions about common objects (from COCO (Lin et al., 2014)). In this context, both the visual modality (seeing the object) and the language modality (understanding the object's name) can independently confirm the object's presence, leading to high informational overlap.

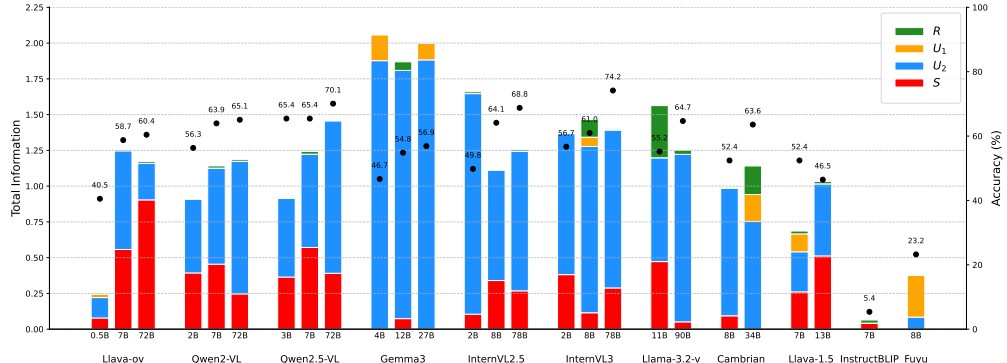

Figure 10: Full information spectrum and accuracy on Reefknot. This plot exemplifies the knowledge-driven regime. Language uniqueness $U_2$ is the overwhelmingly dominant information component for nearly all models, demonstrating that performance is constrained by language-side priors. Even in this environment, the fundamental strategies of model families persist: fusion-centric models (e.g., Llama-3.2-v) still generate noticeably more synergy $S$ than their language-centric counterparts (e.g., Gemma3 and Cambrian).

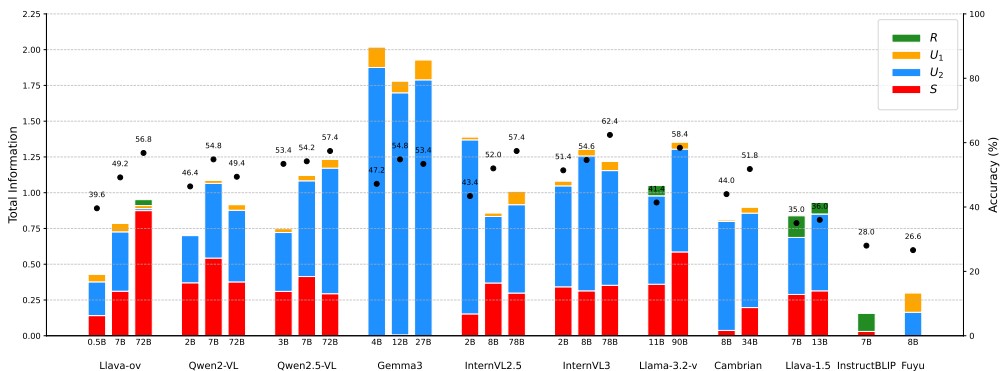

Figure 11: Full information spectrum and accuracy on PMC-VQA. As a specialized, domain-specific task, this dataset clearly exemplifies the knowledge-driven regime. Similar to Reefknot, performance is heavily dominated by language uniqueness $U_2$, confirming that models must rely on internal, language-based knowledge for specialized topics. This result strongly supports our finding that for domain-specific tasks, performance is fundamentally limited by a model's internal, language-based knowledge. The negligible synergy $S$ in many top models highlights the challenge of multimodal fusion when highly specific domain knowledge is required.

## E.2 FULL LAYER-WISE RESULTS ON MMBENCH AND PMC-VQA

This section provides the full, layer-by-layer information spectra for the representative models discussed in our main analysis. These plots show the values for all four PID components $(R, U_1, U_2, S)$ across transformer blocks for each model on both MMBench and PMC-VQA. They provide the detailed evidence for the three-phase layer-wise dynamics shared among different LVLMs.

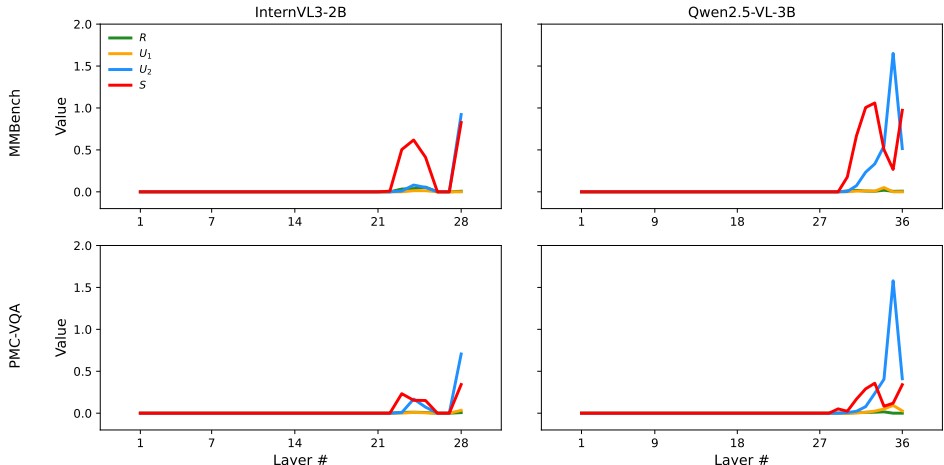

Figure 12: Layer-wise PID dynamics for InternVL3-2B and Qwen2.5-VL-3B. The plot for Qwen2.5-VL-3B clearly illustrates the standard three-phase reasoning process: after information emerges, the later layers show a phase of **representation** building (rising $U_2$), which culminates in a decisive **fusion event** at the final layer (a spike in $S$ and a drop in $U_2$). In contrast, **InternVL3-2B** lacks the final fusion event; its language uniqueness $U_2$ continues to rise into the final layer without the characteristic drop. This may be due to its relatively smaller and shallower LLM, which might not have the capacity for the distinct final-layer fusion seen in other models.

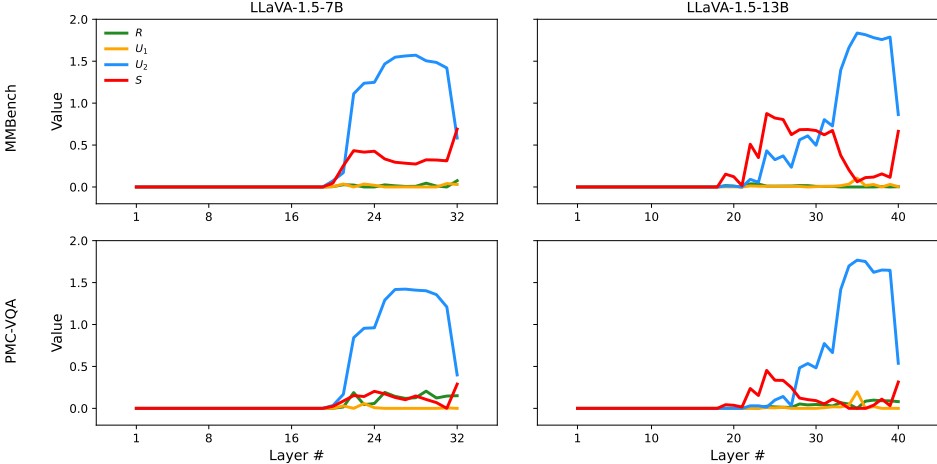

Figure 13: Layer-wise PID dynamics for LLaVA-1.5-7B and LLaVA-1.5-13B. The LLaVA-1.5 family also exhibits the three-phase reasoning process, though with slightly different characteristics. Information emerges in the middle layers of the network. The representation building phase is distinct, with language uniqueness $U_2$ rising to a high plateau while synergy $S$ forms a noticeable "hump," suggesting an ongoing fusion process prior to the final layer. The process concludes with the characteristic fusion event, marked by a drop in $U_2$ and a final spike in $S$ at the output layer. Unlike the other models, LLaVA-1.5 shows some minor, non-zero activity for redundancy $R$ and vision uniqueness $U_1$ in its later layers.

### E.3 Full learning-dynamics results on MMBench and PMC-VQA

This section provides the full information spectra traced across the eight training checkpoints of LLaVA-1.5 (7B and 13B). These plots show the values for all four PID components $(R, U_1, U_2, S)$ on both MMBench and PMC-VQA. They provide the detailed evidence for the learning dynamics and scale-dependent effects identified in Finding 6.

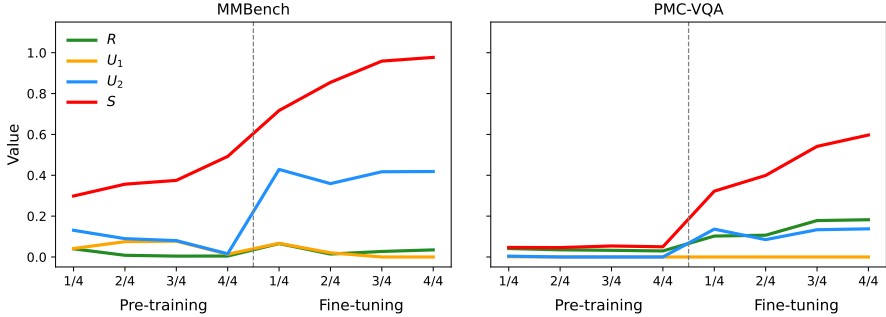

Figure 14: Learning dynamics of LLaVA-1.5-7B. This figure provides the evidence for the first part of Finding 6, showing how smaller models develop fusion. During Stage 1 (Pre-training), all PID components are negligible. Upon commencing Stage 2 (Visual Instruction Tuning), there is a dramatic and sustained increase in synergy $S$, which becomes the dominant information component by the end of training. Language uniqueness $U_2$ also increases but to a much lesser extent, confirming that the 7B model primarily prioritizes developing synergistic inference.

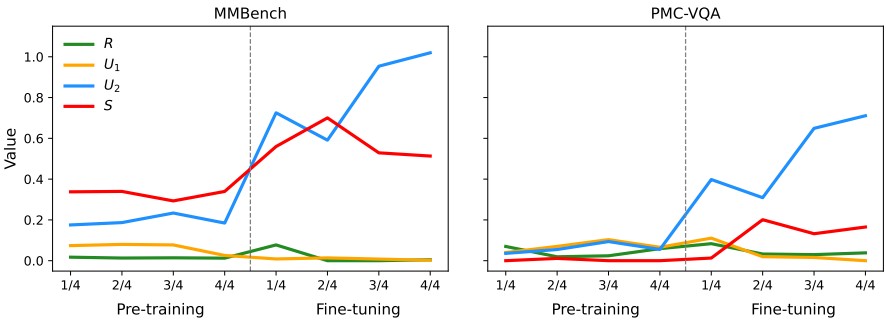

Figure 15: Learning dynamics of LLaVA-1.5-13B. In contrast to the 7B model, this figure illustrates the second part of Finding 6. While PID values are also flat during Stage 1, the larger 13B model exhibits a massive and continuous increase in language uniqueness $U_2$ during Stage 2, becoming by far the dominant information component. Although synergy $S$ also grows, its increase is less pronounced than that of $U_2$, demonstrating that larger models prioritize enhancing their language-side priors during visual instruction tuning.

