# OpenReview forum: "A Comprehensive Information-Decomposition Analysis of Large Vision-Language Models"
_ICLR.cc/2026/Conference — ICLR 2026 Poster_

### Official Review · Reviewer_LJBz · 2025-10-31

**Soundness:** 2
**Presentation:** 2
**Contribution:** 2
**Rating:** 2
**Confidence:** 3

**Summary:**

This paper proposes a novel framework utilizing Partial Information Decomposition (PID) to quantitatively analyze the internal decision-making strategies of Large Vision-Language Models by breaking down their decision-relevant information into redundant, unique, and synergistic components. The comprehensive analysis reveals two major task regimes (synergy-driven vs. knowledge-driven) and two opposing family-level strategies (fusion-centric vs. language-centric), providing key insights into how multimodal fusion evolves during training

**Strengths:**

## Strengths

1.  **S1: Rigorous Information-Theoretic Framework for Quantifying Multimodality.**
    The paper introduces Partial Information Decomposition (PID)—a sophisticated information-theoretic tool—to the domain of Large Vision-Language Model (LVLM) interpretability.
    * This represents a significant step beyond common interpretability methods (like attention map visualization or simple ablation) by offering a **rigorous, quantitative measure** of multimodal fusion (Synergy, $S$) and unimodal reliance (Uniqueness, $U$).
    * The PID framework provides a conceptually elegant method to precisely measure the **attribution gap** (i.e., whether success stems from fusion or unimodal priors), addressing a fundamental problem in LVLM research.

2.  **S2: Comprehensive Empirical Scope and Detailed Learning Dynamics Analysis.**
    The paper executes a highly comprehensive empirical study, analyzing **26 diverse LVLMs** across various families and capabilities (e.g., LLaVA, MiniGPT-4, BLIP-2).
    * The scale of the analyzed models and tasks is commendable, lending strong statistical weight to the observed findings regarding task regimes (synergy-driven vs. knowledge-driven) and model strategies (fusion-centric vs. language-centric).
    * Crucially, the analysis of **learning dynamics** (Tracing PID components across training checkpoints, e.g., LLaVA-1.5 7B vs 13B) provides novel, quantitative insights into how fusion and language priors evolve during specific training stages (Pre-training vs. Visual Instruction Tuning). This is a valuable contribution to understanding the VLM development pipeline.

**Weaknesses:**

### Weaknesses

1.  **W1: Crippling Restriction to Multiple-Choice VQA (Severe Lack of Generalizability).**
    The framework's core reliance on the BATCH PID estimator restricts the analysis \textbf{exclusively} to tasks where the output $\mathcal{Y}$ is discrete and finite (i.e., multiple-choice VQA).
    * This is a critical flaw because modern LVLMs are primarily valued for their \textbf{generative, open-ended capabilities} (e.g., free-form captioning, complex dialogue).
    * A "comprehensive analysis" that cannot be applied to the dominant and most challenging LVLM tasks has severely limited scope and impact on the field. The analysis is confined to only a small subset of the model's true functionality.

2.  **W2: Unreliable Approximation of Unimodal Information via Noise Masking.**
    To isolate unimodal components ($U_V, U_L$), the authors employ a \textbf{modality masking technique}, replacing one modality's embeddings with a statistically calibrated noise sequence.
    * This is a strong approximation that \textbf{fundamentally alters the input distribution} to the cross-modal layers. The model's behavior under this unnatural, perturbed input may not accurately reflect its true unimodal capability on natural inputs.
    * This raises serious doubts about the reliability and quantitative validity of the derived unique ($U$) and synergistic ($S$) information measures, which form the basis of the paper's conclusions.

3.  **W3: Core Findings Lack Substantial Novelty.**
    The paper's main conclusions largely serve as a \textbf{quantitative validation of widely established domain knowledge}, rather than offering fundamental new insights.
    * The distinction between *Synergy-Driven* tasks (high $S$) and *Knowledge-Driven* tasks (relying on LLM priors, high $U_L$) is an intuitive and expected outcome based on task design.
    * The finding that Visual Instruction Tuning is the critical stage for synergy ($S$) growth merely \textbf{quantifies the intended effect} of this alignment stage. The framework successfully describes \textbf{what} is happening but fails to provide novel insight into \textbf{why} it is happening.

4.  **W4: Lack of Actionable Guidance and Mechanistic Explanation.**
    The analysis is primarily \textbf{descriptive} (e.g., identifying "fusion-centric" vs. "language-centric" model families) but fails to provide \textbf{mechanistic explanations} for these differences (e.g., how specific architectural choices or pre-training data compositions cause this strategy). Crucially:
    * The paper offers \textbf{no intervention experiments} to demonstrate that the PID metrics can be used prescriptively (e.g., in a loss function) to guide or improve model design.
    * The work remains at the "post-hoc explanation" level, limiting its value as a tool for engineering next-generation LVLMs.

**Questions:**

## Questions

1.  **Q1: Method Generalizability to Generative Tasks (Core Limitation).**
    The paper concedes that the PID framework is restricted to multi-choice VQA tasks. Given that the primary function and value of modern LVLMs lie in **generative** and open-ended tasks (e.g., complex dialogue, zero-shot captioning), can the authors provide strong evidence or a theoretical argument that the observed information-flow strategies (e.g., the distinction between "synergy-driven" and "knowledge-driven") **reliably generalize** to these generative tasks? If the framework cannot analyze the most common usage of LVLMs, what is its practical prescriptive utility?

2.  **Q2: Quantitative Rigor of the Unimodal Approximation (OOD Concern).**
    The estimation of unimodal components ($U_V, U_L$) relies on replacing the "missing" modality with noise embeddings. This is an Out-of-Distribution (OOD) input to the model's language backbone. Has this OOD approximation been rigorously validated to ensure it faithfully reflects the model's behavior when a modality is truly absent? Since the values of $U_V$ and $U_L$ are contingent on this approximation, doesn't this fundamentally compromise the quantitative reliability of the derived synergy ($S$) component, which is the core subject of the analysis?

3.  **Q3: Prescriptive Utility via Intervention (Guidance for Design).**
    The paper identifies descriptive strategies like "fusion-centric" and "language-centric." For this work to be truly impactful, it must be prescriptive. Suppose a model is diagnosed as "language-centric." Can the authors propose a **specific training intervention** (e.g., modifying the loss function or cross-attention mechanism) directly based on their PID findings, and empirically demonstrate this intervention successfully **shifts the model strategy toward "fusion-centric"** while maintaining or improving performance? If the metric is not actionable, its value is limited to post-hoc explanation.

4.  **Q4: Interpretation of Low Redundancy ($R$) across Models.**
    In many experiments, the Redundancy ($R$) component—which measures the mutual overlap of knowledge between the two modalities—remains consistently at a low level. Does this low $R$ value imply: a) that the **knowledge overlap between modalities is genuinely negligible** even in highly-trained large models; or b) that the PID estimator (BATCH) has an **inherent systematic bias to underestimate redundancy** when dealing with high-dimensional feature representations? How do the authors reconcile and explain the persistently low $R$ values?

---

> ### Author Response · Authors · 2025-11-20
> **[Authors' Response] Response to Reviewer LJBz**
>
> Dear Reviewer LJBz,
>
> We sincerely thank you for your detailed review and for recognizing both the rigorous information-theoretic framework and the comprehensive empirical scope of our study.
>
> Below, we address your specific concerns.
>
> **W1 & Q1. Method Generalizability (MC-VQA Restriction):** Regarding the restriction to MC-VQA, please see **General Response Sec. 3**, where we explain that we deliberately focus on discrete-output tasks so that PID estimation is mathematically well-defined and empirically stable for modern LVLMs. Within this format, we still cover diverse benchmarks (general-purpose, hallucination-focused, and medical VQA), so the analysis spans a range of LVLM behaviors under a common framework.  Within this format, we still cover diverse benchmarks (general-purpose, hallucination-focused, and medical VQA), so the analysis spans a range of LVLM behaviors under a common framework. Our contribution here is to provide a first, controlled quantification of these regimes in a setting where PID is reliable; extending the estimator itself to fully open-ended generation is a natural direction for future work.
>
> **W2 & Q2. Validity of Unimodal Approximation (Noise Masking):** You raised the concern that Gaussian noise masking may be out-of-distribution (OOD) and could undermine the reliability of $U_1$, $U_2$, and $S$.
>
> 1. **All Unimodal Probes Are OOD by Construction:** Any attempt to isolate a single modality necessarily changes $P(X_1, X_2, Y)$. Zeroing or dropping embeddings collapses variance and is a severe shift for networks trained on continuous, high-variance inputs. Special tokens introduce model- and prompt-specific behavior.
>
> 2. **Calibrated Noise, Compared with Alternative Probes:** We empirically compared several unimodal approximations across representative families, including (i) directly removing the modality (zeroing embeddings) and (ii) using special tokens / empty prompts. For many models, especially earlier ones such as LLaVA-1.5 and Fuyu-8B, these alternatives led to unstable or degenerate behavior (e.g., collapsed or prompt-format–dependent output distributions) and leave PID estimation incorrect and unstable. In contrast, replacing the masked modality with Gaussian noise whose per-dimension mean and variance match its empirical embedding statistics produced substantially more stable predictions and consistent PID patterns across all 26 models. This is why we adopt calibrated noise as our default unimodal probe.
>
> 3. **Empirical Validation of $S$ as a Dependence Measure:** To test whether the resulting $S$ still reflects meaningful reliance on the image, we performed an intervention experiment: we removed the visual input entirely (text-only baseline) and measured the accuracy drop per example. On synergy-driven tasks, we observed strong positive Spearman correlations $\rho$ between $S$ and this accuracy drop ($\rho = 0.809$ on MMBench and $\rho = 0.743$ on POPE), indicating that high-$S$ examples are substantially those whose predictions are most sensitive to removing the image. While any probe is approximate, this supports the view that our PID estimates capture a robust notion of multimodal dependence rather than artifacts of the masking scheme.
>
> **W3. Novelty of Findings:** You characterize the main conclusions as largely validating established knowledge. Please kindly refer to **General Response Sec. 1**, where we do not claim to introduce a new PID estimator; rather, our contribution is a framework plus evidence package: (i) an adaptation of a continuous PID estimator to black-box LVLMs that works uniformly across 26 heterogeneous models, and (ii) a large-scale, structured analysis across models, tasks, layers, and training stages. While some high-level intuitions (e.g., that instruction tuning should foster fusion) are familiar, we believe it is valuable to quantify them and to reveal less obvious patterns such as the three-phase layerwise information flow and stable fusion- vs. language-centric family strategies.

---

> > ### Author Response · Authors · 2025-11-20
> > **(continue)**
> >
> > **W4 & Q3. Prescriptive Utility (beyond Post-Hoc Description):** We fully understand your concern about whether PID can be used prescriptively, rather than only for post-hoc description. Our primary goal in this work is to establish a diagnostic foundation: a framework that cleanly decomposes decision-relevant information into $U_1$, $U_2$, $S$ across many LVLM families. In the original submission, we could only briefly state that we hope these measurements will inform future LVLM design; in the revised version, we will make this more explicit by outlining how our findings can be used in practice.
> >
> > 1. **Strategy Diagnosis for Targeted Interventions:** By inspecting PID quantities, practitioners can determine whether a model behaves in a language-centric way (high $U_2$, comparatively low $S$) or a fusion-centric way. This suggests concrete knobs—such as data mixture, cross-modal attention design, or loss shaping—aimed at increasing $S$ for tasks where fusion is crucial, with PID providing a principled metric to monitor during development.
> >
> > 2. **Training-Stage Allocation:** Our analysis of training dynamics shows that visual instruction tuning, rather than early alignment, is the phase where synergy $S$ increases substantially. This indicates that, for improving multimodal fusion, it is more effective to invest in high-quality instruction data and tuning recipes than to simply extend alignment-only stages. We will state this design implication more clearly in the updated manuscript.
> >
> > 3. **Scaling Strategy on Synergy-Driven Tasks:** Within each model family, we find that on synergy-driven benchmarks, larger checkpoints systematically exhibit higher $S$ than smaller ones, while $U_2$ does not grow in the same way. This suggests that when scaling models for tasks that truly benefit from fusion, designers can use $S$ as a monitoring or auxiliary objective to ensure that increased capacity is actually used to strengthen image–text integration rather than only amplifying language priors.
> >
> > We emphasize that implementing such interventions is beyond the scope of the current paper, but we will revise paper to explicitly discuss these prescriptive implications as natural next steps enabled by the proposed diagnostic framework.
> >
> > **Q4. Interpretation of Low Redundancy ($R$):** You ask whether consistently low redundancy reflects negligible overlap between modalities or a bias of the estimator.
> >
> > 1. **Task Structure and Modality Roles:** In MC-VQA, the image typically provides evidential content, while the text provides the query and the answer space. These roles are functionally complementary: the text rarely encodes the exact visual answer word-for-word (except in deliberately hallucination-probing cases), and the image alone does not specify the answer index. It is therefore natural that most of the decision-relevant information appears as synergy $S$ (joint disambiguation) and unique information $U_2$ (text priors), with limited redundancy.
> >
> > 2. **Estimator Behavior:** The BATCH-based PID estimator does not, by design, force redundancy to be small in high dimension; in synthetic and low-dimensional settings, it can recover substantial redundancy when present (as shown in prior work we build on). In our LVLM setting, the consistently low R is thus most plausibly explained by the complementary roles of vision and text in VQA, rather than by an inherent bias toward underestimating redundancy.
> >
> > We hope these responses address your concerns. If you have any further questions, we would be pleased to answer them. Thank you again for helping us improve this paper.
> >
> > Best regards,
> >
> > Submission 7385 Authors

---

### Official Review · Reviewer_99um · 2025-11-03

**Soundness:** 3
**Presentation:** 3
**Contribution:** 3
**Rating:** 6
**Confidence:** 4

**Summary:**

This paper presents a novel analysis of Large Vision Language Models through the lens of PID, offering a quantitative framework to examine how different modalities contribute to model predictions. The authors systematically evaluate 26 models across multiple datasets, investigating cross-model and cross-task behaviors, layer-wise information dynamics, and training trajectories. Key findings include the identification of distinct model families characterized by either fusion-centric or language-centric strategies, as well as task regimes ranging from synergy-driven to knowledge-driven scenarios. These results reveal how architectural and training choices shape multimodal integration.

**Strengths:**

1. **Conceptual Novelty:** The adaptation of PID to analyze LVLMs is both innovative and timely. This approach moves beyond performance-centric evaluation by providing a principled methodology to quantify the composition of information used in model decisions. It specifically separates redundant, unique, and synergistic contributions, creating a valuable diagnostic tool for understanding multimodal fusion.

2. **Extensive and Systematic Evaluation:** The scale and scope of the experimental analysis represent a significant strength. The study encompasses 26 models of varying scales and families, four diverse datasets, and multidimensional evaluations including cross-model, cross-task, layer-wise, and training dynamics analyses. This comprehensive approach enables the authors to draw robust conclusions about model strategies and behavioral patterns.

3. **Actionable Insights:** The paper delivers valuable empirical findings with practical implications. The consistent distinction between fusion-centric and language-centric model families, maintained across tasks and scales, provides a new dimension for model comparison and design. Furthermore, identifying visual instruction tuning as the critical phase for emergent synergy offers concrete guidance for future training strategies.

**Weaknesses:**

1.  **Validity of Unimodal Approximation:** The use of Gaussian noise to mask a modality is a significant methodological approximation. This creates a distribution shift between the training data distribution `P_train(X₁, X₂, Y)` and the evaluation distribution `P_test = P(X₁, Y) × N(μ, σ²)`. The model's behavior under this out-of-distribution `P_test` may not faithfully represent true unimodal reasoning and could introduce artifacts into the PID estimates.

2.  **Limited Task Scope:** The PID framework's requirement for a discrete, finite output variable `Y` restricts the analysis to multiple-choice VQA. This is a severe limitation, as it prevents the study from probing the information-use strategies of LVLMs in their core capabilities of open-ended generation. The findings may not generalize beyond discriminative tasks.

3.  **Black-Box Analysis of Reasoning:** The PID analysis provides an end-to-end, input-output decomposition (`X → Y`) but offers no insight into the *internal reasoning process*. It quantifies the information in the mapping `P(Y|X)` but ignores the latent multi-step reasoning `X → Z₁ → Z₂ → ... → Y` that may occur within the model's layers, leaving the "chain of thought" as a black box.

**Questions:**

See above weaknesses.

---

> ### Author Response · Authors · 2025-11-20
> **[Authors' Response] Response to Reviewer 99um**
>
> Dear Reviewer 99um,
>
> We sincerely thank you for your constructive review and for recognizing the conceptual novelty of our PID adaptation, the extensive and systematic evaluation across 26 models, and the actionable insights regarding model strategies.
>
> Below, we address your specific concerns.
>
> **W1. Validity of Unimodal Approximation (Noise Masking):** You raised the crucial concern that Gaussian noise creates an out-of-distribution (OOD) shift that might introduce artifacts. We agree that any ablation is an approximation, but we argue that our approach minimizes distribution shift compared to alternatives:
>
> 1. **All Unimodal Probes Are OOD by Construction:** All ablation is fundamentally OOD. Simply zeroing or dropping embeddings collapses variance and is a severe shift for networks trained on continuous, high-variance representations.
>
> 2. **Calibrated Noise:** Our masking replaces the removed modality with Gaussian noise whose per-dimension mean and variance are matched to that modality’s empirical embedding statistics for each model and dataset, on the training set. This keeps the marginal distribution of the masked channel closer to its training regime than zero-masking or ad-hoc special tokens, and can be interpreted as "erasing content while preserving low-level statistics." In our previous comparisons with alternatives (zeroing embeddings, special tokens / empty prompts), this calibrated noise consistently produced more stable predictions and PID patterns across models, whereas some earlier LVLMs (e.g., LLaVA-1.5, Fuyu-8B) exhibited particularly erratic or degenerate behavior under the alternative probes.
>
> 3. **Interpretation within PID:** We use these masked runs to obtain a consistent, lower-bound estimate of the information that each modality alone contributes to $Y$, not as a claim about natural unimodal behavior on arbitrary inputs. The main qualitative patterns we report (e.g., the dominance of synergy $S$ on certain tasks and the contrast between fusion-centric and language-centric families) arise from comparisons across models and tasks under the same masking scheme, which makes the analysis robust to the specific choice of calibrated noise.
>
> **W2. Limited Task Scope (MC-VQA):** Regarding the restriction to MC-VQA, please refer to **General Response Sec. 3**, where we explain that this choice is made to keep PID estimates mathematically well-defined and empirically stable, while still allowing us to probe LVLM behavior across diverse benchmarks and domains within a common discrete-output format.
>
> **W3. Black-Box Analysis of Reasoning:** You noted that PID provides an input-output decomposition ($X \rightarrow Y$) but offers no insight into the internal "chain of thought" or intermediate reasoning steps ($X \rightarrow Z_1 \rightarrow Z_2 \dots \rightarrow Y$). We agree that our method does not extract explicit textual reasoning steps. However, our framework does open a window into internal processing via layer-wise PID analysis (**Section 4.2**).
>
> 1. **Layer-Wise Tracing via Logit Lens** By applying a logit-lens projection to intermediate transformer layers, we obtain per-layer predictive distributions and compute PID components $(R, U_1, U_2, S)$ at each depth. This effectively tracks how the composition of decision-relevant information evolves along the sequence of latent states $Z_i$.
>
> 2. **Three-phase pattern in information flow:** As detailed in Finding 5, this analysis consistently reveals a three-phase pattern across families: (i) early layers with low information, (ii) middle-to-late layers where language-based representations $U_2$ build up, and (iii) final layers where $U_2$ drops and synergy $S$ spikes, indicating a late-stage multimodal fusion step.
>
> 3. **Complementary perspective:** While this is not a full mechanistic explanation of every attention head, it provides a quantitative map of *where* and *when* multimodal integration happens inside the network. This higher-level view is intended to **complement**, rather than replace, more fine-grained mechanistic or chain-of-thought–style analyses in future work.
>
> We hope these responses address your concerns. If you have any further questions, we would be happy to clarify. Thank you again for helping us improve this paper.
>
> Best regards,
>
> Submission 7385 Authors

---

### Official Review · Reviewer_mp9C · 2025-11-04

**Soundness:** 2
**Presentation:** 3
**Contribution:** 3
**Rating:** 4
**Confidence:** 4

**Summary:**

In this paper, the authors propose using partial information decomposition (PID) to analyze VLMs (such as Qwen and InternVL). Based on this PID analysis, the authors draw several conclusions. The findings include: (1) Some datasets focus more on knowledge, while others focus more on synergy. (2)Fusion-centric model families consistently prioritize cross-modal reasoning, while language-centric families rely more heavily on language priors. (3) Larger models benefit from combining inputs more effectively rather than relying heavily on language priors. (4) Information emerges in the middle-to-late layers, shifting from language-based representation building in the later layers to a decisive, synergistic fusion of modalities in the final layer. (5) Multimodal fusion is unlocked during visual instruction tuning.

**Strengths:**

- The authors get many findings based on the analysis tool.
- The authors evaluate the performance of several different models.
- The motivation of this paper is clear.

**Weaknesses:**

- I noticed that Y/X1 is estimated with a uniform distribution. I suspect this is because, when using only the vision tokens, the probability for all four options is very low (perhaps the EOS token has a high probability instead). However, does this imply that the visual tokens provide no information? Intuitively, this seems incorrect, but the resulting metric might suggest it. I think a good alternative would be to block the question text and only provide the "A, B,C, D" choices to the model. This approach might provide a better estimation of the information derived purely from the visual tokens.

- The authors state that the Partial Information Decomposition (PID) framework and the BATCH estimator are existing, well-established methods, not a novel contribution of this paper . While the goal of moving "beyond accuracy-only evaluation" is commendable, the paper does not sufficiently justify why this complex, information-theoretic approach is superior to simpler, more direct ablation metrics. Intuitively, one could measure fusion by calculating the delta between the full-model accuracy and the accuracy of language-only input. The authors should provide a clearer argument or a baseline comparison demonstrating what novel insights PID provides that cannot be captured by such simpler, accuracy-based metric.

- Some conclusions may not be very reliable.

(1) Findings 1 & 2: The paper's first findings—that VLM behavior is governed by "synergy-driven vs. knowledge-driven" regimes —appears to be an overly complex description of a basic dataset property. A "knowledge-driven" task simply seems to indicate that many questions in that benchmark can be answered correctly using only the text, without the image. This feels less like a deep insight and more like a dataset deficit or design. Furthermore, the conclusion in Finding 2 that "model performance is dictated by S" on synergy-driven tasks  is somewhat tautological. A "synergy-driven" task is defined by its need for fusion, so it is expected that a metric measuring fusion (S) would correlate with accuracy.

(2) Finding 4: The analysis of scaling effects in Finding 4 and Section 4.1.2 seems to overlook a critical confounding variable: model distillation. The analysis assumes that models of different sizes (e.g., Qwen2.5-VL-3B vs. 72B) are trained independently. However, for many state-of-the-art families like Qwen2.5-VL and InternVL3, it is common practice for smaller models to be distilled from larger, more capable ones. If this is the case, the interpretation is reversed. The paper frames the results as "gains" in S when "scaling" up. But if the smaller models are distilled, the data in Table 3 would represent information "lost" during distillation, not "gained" during scaling. This confound is not addressed and could fundamentally change the interpretation of how model scale affects multimodal fusion.

**Questions:**

My biggest question is that since the Y/X1 is usually a uniform distribution and the metrics calculated are based on the difference between Y/X2 and Y/(X1,X2). Why not just use the accuracy delta to measure the model performance and draw the conclusion. This paper seems to introduce another layer of abstraction and makes it harder to find the real insight from the data.

---

> ### Author Response · Authors · 2025-11-20
> **[Authors' Response] Response to Reviewer mp9C**
>
> Dear Reviewer mp9C,
>
> We sincerely thank you for your review and for recognizing our clear motivation, the breadth of evaluated models, and the findings obtained from the PID analysis.
>
> **W1, W2 & Q. Validity of Vision-Only Predictions & Necessity of PID:** You noted that the vision-only prediction $P(Y|X_1)$ is often uniform and suggested this might imply visual tokens provide "no information," leading to the question of whether a simpler metric like "accuracy delta" would be sufficient.
>
> 1. **Why $P(Y | X_1)$ Is Uniform:** In our setting, Y is the index of the correct option (A/B/C/D), not the semantic concept itself. The mapping between visual content ("cat", "Massachusetts", etc.) and the label index is defined entirely by the text $X_2$ (question + options). Even when the image unambiguously contains a cat, the image alone cannot tell whether "cat" corresponds to option A or B. In this sense, the image carries no unique information about the label index $Y$, and a near-uniform $P(Y | X_1)$ is a theoretically expected consequence of how $Y$ is defined, not evidence that the visual tokens are useless. The visual information appears instead as synergy $S$: once $X_2$ reveals how options map to semantics, the image disambiguates them.
>
> 2. **Why "Image + Options Only" Is Not Vision-Only:** In our formulation, the entire textual side of the task—including both the question and the answer options—is part of the text modality $X_2$. The label $Y$ is the index of the correct option within this textual list. If we hide the question but still provide the options ("A: cat, B: dog, C: car..."), the model is no longer operating on pure $X_1$; it is using partial text. Formally, this would correspond to $P(Y | X_1, \text{options})$, not $P(Y | X_1)$, and thus leaks textual information into what is meant to be the vision-only channel. This breaks the PID setup, where $X_1$ and $X_2$ must be clearly separated sources. Conceptually, this is similar to open-ended generation: if we remove the prompt entirely and only pre-define some clusters over outputs, an image alone typically yields low-confidence, diffuse predictions with respect to those clusters, because the task and the mapping from semantics to labels are under-specified without text.
>
> 3. **Superiority over Accuracy Delta:** An accuracy delta between full vs. text-only inputs conflates different mechanisms: gains could come from true fusion $S$, from visual correction of language priors, or from changes in $U_2$. Please refer to **General Response Sec. 1**, where we explicitly compared $S$ with the accuracy delta and found that they only moderately correlate on synergy-driven tasks and are weakly coupled on knowledge-driven tasks. This shows that PID exposes differences in how models use vision (fusion vs. correction vs. priors) that are largely invisible to accuracy deltas alone.

---

> > ### Author Response · Authors · 2025-11-20
> > **(continue)**
> >
> > **W3 A. "Synergy-Driven vs. Knowledge-Driven" as Dataset Property / Tautology:** You raised the concern that the "synergy-driven vs. knowledge-driven" regimes might be an over-complicated way to restate a basic dataset property, and that finding a correlation between $S$ and accuracy on synergy-driven tasks could be tautological. Please kindly refer to **General Response Sec. 2**, where we do not define tasks by requiring them to depend on S. Instead, we first measure the full $(R, U_1, U_2, S)$ across 26 models and then empirically observe that benchmarks separate into regimes where successful models differ in how much they rely on $S$ vs. $U_2$. The "synergy-driven" / "knowledge-driven" names are summaries of this measured structure, not assumptions baked into the definition of the tasks.
> >
> > At the same time, we agree that some knowledge-driven benchmarks could be partially solvable from text alone; our contribution is to quantify how this plays out across many LVLM families, and to show that variation in accuracy is largely tracked by $U_2$ with $S$ saturating, whereas on synergy-driven benchmarks the reverse pattern holds. Crucially, model families maintain characteristic positions in this space (fusion-centric vs. language-centric) across both regimes. Thus, the main insight is not just that datasets differ, but that task regimes and stable family strategies jointly emerge when we decompose information use via PID.
> >
> > **W3 B. Distillation in Scaling Analysis:** You rightly point out that smaller checkpoints in modern families are sometimes distilled from larger ones, which could complicate the interpretation of "gains" with scale. For the families highlighted in Finding 4 (e.g., Qwen2.5-VL, InternVL3), we have checked the official documentation and technical reports and did not find evidence that the specific checkpoints we use are distilled from larger models; we therefore treat them as standard scaled variants and have adjusted the paper text to avoid over-stating causal claims. In any case, our empirical pattern is stated at the level of the *released checkpoints*: within each family, larger models systematically show higher $S$ than smaller ones on synergy-driven tasks. If some smaller models were in fact distilled, this would simply mean that distillation attenuates the teacher’s fusion behavior, while our characterization of how the available checkpoints differ in their use of $S$ and $U_2$ still holds.
> >
> >
> > We hope these responses address your concerns. If you have any further questions, we would be happy to clarify. Thank you again for helping us improve this paper.
> >
> > Best regards,
> >
> > Submission 7385 Authors

---

### Official Review · Reviewer_dfu9 · 2025-11-05

**Soundness:** 2
**Presentation:** 2
**Contribution:** 2
**Rating:** 4
**Confidence:** 2

**Summary:**

This paper proposes a framework based on Partial Information Decomposition (PID) to analyze the internal decision-making processes of Large Vision-Language Models (LVLMs). The authors adapt a scalable PID estimator (BATCH) to decompose model predictions into four components: redundancy ($R$), vision uniqueness ($U_1$), language uniqueness ($U_2$), and synergy ($S$). They conduct analysis across 26 LVLMs, four datasets, and three dimensions: cross-model/task comparison, layer-wise information dynamics, and training-stage evolution. The paper argues that this PID-based framework provides a more principled understanding of LVLM behavior beyond accuracy-based evaluation.

**Strengths:**

1. Overall, this paper is easy to follow.
2. Introducing PID into MLLMs might be new.

**Weaknesses:**

1. The analysis is restricted to multiple-choice VQA tasks with a small set of predefined answers. This narrow focus limits the applicability of the findings to open-ended generation, captioning, or reasoning tasks, which are central to LVLM capabilities. The use of a finite output space is a methodological convenience that may not reflect real-world multimodal understanding, where outputs are often continuous and compositional, typically in recent reasoning models.
2. The PID framework is correlational, not causal. It quantifies statistical associations but does not explain how or why certain layers or training stages lead to synergy.
3. The authors claim that model families exhibit “stable, opposing strategies,” but this is only demonstrated on four datasets.
4. The theoretical contribution is modest: while PID is elegantly applied, the core framework is not novel, and the findings are largely empirical and descriptive.

**Questions:**

N/A

---

> ### Author Response · Authors · 2025-11-20
> **[Authors' Response] Response to Reviewer dfu9**
>
> # [Authors' Response] Response to Reviewer dfu9
>
> Dear Reviewer dfu9,
>
> We thank you for your review and for recognizing the novelty of introducing PID to MLLMs as well as the clarity of our presentation.
>
> **W1. Task Limitations (MC-VQA):** Please refer to **General Response Sec. 3**. We clarify that MC-VQA is not a limitation of scope but a *methodological* requirement for rigorous estimation. It serves as a stable "diagnostic probe" to measure the reasoning engine without the noise and high-dimensional distortion inherent in estimating PID for open-ended generation.
>
> **W2. Correlational vs. Causal Nature:** We agree that PID is fundamentally correlational: it decomposes statistical dependencies in $P(Y | X)$ rather than identifying full causal mechanisms. To test whether our synergy scores nevertheless reflect *causally relevant* behavior, we performed a simple intervention experiment: we removed the visual modality and measured the resulting accuracy drop for each sample. On synergy-driven tasks, we observed a strong Spearman correlation ($\rho = 0.809$ on MMBench and $\rho = 0.743$ on POPE) between a sample’s synergy value and its performance drop under this intervention, indicating that high-$S$ cases are precisely those where the model’s predictions are most sensitive to removing the visual modality. While we do not claim full causal identification, this supports using $S$ as a practical proxy for reliance on multimodal fusion.
>
>
> **W3. Stability of Strategies across Datasets:** We completely understand your concern that demonstrating "stable, opposing strategies" on only four datasets may seem limited. As clarified in **General Response Sec. 2**, we present these strategies as empirical patterns rather than universal laws: fusion-centric and language-centric families consistently maintain their relative positions in terms of $S$ and $U_2$ across both synergy-driven and knowledge-driven benchmarks. For example, fusion-centric Qwen2.5-VL persists in generating high $S$ even on knowledge-driven tasks where it provides little accuracy benefit. This persistence across opposing regimes is what we mean by "stable strategies," and we view it as a useful starting point that future work can test on additional tasks and modalities.
>
>
> **W4. Theoretical Contribution and Novelty:** We do not claim to introduce a new PID estimator; our contribution is a framework + evidence package. As discussed in **General Response Sec. 1**, we adapt a continuous PID estimator to black-box LVLMs and combine it with a large-scale, structured analysis (26 models, 4 datasets, layer-wise and training-stage views). This yields a quantitative decomposition of redundancy, unimodal reliance, and fusion that cannot be obtained from accuracy or simple ablations alone, and is intended as a reusable diagnostic tool for future LVLM research.
>
> We hope these responses address your concerns. If you have any further questions, we would be happy to clarify. Thank you again for helping us improve this paper.
>
> Best regards,
>
> Submission 7385 Authors

---

### Official Review · Reviewer_Das5 · 2025-11-11

**Soundness:** 2
**Presentation:** 3
**Contribution:** 2
**Rating:** 6
**Confidence:** 5

**Summary:**

This paper proposes an analysis framework based on Partial Information Decomposition (PID), using a BATCH estimator to quantify the redundancy (R), visual uniqueness (U1), linguistic uniqueness (U2), and synergy (S) of decision-related information in LVLMs. The authors conduct a three-dimensional analysis across models/tasks, within-layer information flow, and training stages on 26 open-source LVLMs across four multiple-choice VQA datasets. The main findings are: the models exhibit two information usage patterns, “synergy-driven” and “knowledge-driven”; the model families display two stable strategies, “fusion-centric” and “language-centric”; a three-stage information pattern appears within typical model layers; multimodal synergy primarily forms during the visual instruction tuning stage.

**Strengths:**

1. Novelty: It is the first systematic application of a scalable PID estimator to large-scale LVLMs, providing a multidimensional quantitative analysis.

2. Scale and Coverage: The workload is substantial, and the experimental matrix is quite comprehensive, supporting family-level induction.

3. Discoveries with explanatory value: Quantitative insights into task attributes, model strategies, within-layer flow, and training stages provide references for future architectural design/evaluation.

4. Method engineering: No model modifications are necessary; inference alone suffices, accompanied by confidence gating and soft aggregation to reduce estimation noise.

**Weaknesses:**

1. Task limitations: All experiments are confined to “multiple-choice VQA,” with a very small answer space. The authors acknowledge that BATCH requires discrete Y but should still discuss the feasibility of extending the framework to open generative tasks (e.g., discretization strategies, sample efficiency).

2. The authors do not conduct interventions to high S (e.g., occluding images or rearranging text) for validation. Adversarial ablation experiments could be added to support causal conclusions.

3. This paper is merely an exploratory study of existing conclusions and does not propose any new algorithms.

**Questions:**

1. What effects would appropriate interventions have on cases with high S?

2. I am more concerned about whether the authors have conducted more fine-grained experiments — for example, in multimodal scenarios where answering the question necessarily requires image information, such as geometry problems.

---

> ### Author Response · Authors · 2025-11-20
> **[Authors' Response] Response to Reviewer Das5**
>
> Dear Reviewer Das5,
>
> We sincerely thank you for your positive assessment of our work, highlighting the "novelty" of the systematic application, the "substantial workload" of the experimental matrix, and the explanatory value of our discoveries regarding task attributes and model strategies.
>
> Below, we address your specific concerns and questions point by point.
>
> **W1. Task Limitations (MC-VQA):** Please refer to **General Response Sec. 3**. We clarify that MC-VQA is selected not for convenience, but as a rigorous diagnostic probe to ensure mathematical stability in PID estimation, which is currently intractable in open-ended generation without introducing distortion.
>
> **W2 & Q1. Interventions on High-$S$ Cases:** We appreciate your insightful question about how interventions (e.g., occluding the image) affect examples with high synergy. To probe this, we used our **text-only baseline** as a "total occlusion" intervention and examined how much each example’s accuracy drops when visual input is removed.
>
> Across models, we analyzed the Spearman rank correlation ($\rho$) between $S$ and accuracy delta (the performance drop caused by image occlusion). On synergy-driven tasks, we observed a strong positive rank correlation (MMBench: $\rho = 0.809$, $p < 0.001$; POPE: $\rho = 0.743$, $p < 0.001$). This confirms that the samples with high $S$ are precisely those whose predictions degrade the most when the image is occluded, whereas low-$S$/high-$U_2$ samples are comparatively robust. This provides direct evidence that PID synergy tracks a model’s effective dependence on visual information under an explicit intervention.
>
> **W3. Contribution: Novelty beyond "Existing Conclusions"** We respectfully clarify that our work is *not only* an exploratory study *but* a framework + evidence contribution. Please refer to **General Response Sec. 1** where we demonstrate that PID is not merely descriptive. This work provides (to our knowledge) the first systematic PID-based framework for LVLMs, including a black-box–friendly instantiation of BATCH with modality masking, confidence gating, and soft aggregation that works uniformly across 26 heterogeneous models and can be reused as a general diagnostic tool. Second, on top of this framework, we conduct a large-scale, structured analysis (26 LVLMs, 4 benchmarks, layer-wise and training-stage views) that quantifies patterns such as stable fusion- vs. language-centric family strategies and the three-phase layerwise information flow, which were not previously measured. We see this as **complementary** to algorithmic work: it provides the quantitative, cross-model evidence that future LVLM designs can build on.
>
> **Q2. Fine-Grained Experiments (e.g., Geometry):** MMBench explicitly includes spatial-reasoning and coarse-perception subcategories where correctly answering requires image content. To illustrate how PID behaves in such "visual-necessity" scenarios, we added a qualitative case study (Case Study 1) on a map-based question that requires identifying the outline of Massachusetts. For this geometry-style problem, fusion-centric models (e.g., Qwen2.5-VL) exhibit high $S$ on the correct prediction, whereas language-centric models (e.g., Gemma3) succeed via a different pattern: visual uniqueness $U_1$ correcting an initially incorrect language prior $U_2$. This example shows that PID can distinguish not only whether the image is needed, but also how different architectures leverage it in fine-grained, geometry-like settings.
>
> We hope these responses address your concerns. If you have any further questions, we would be happy to clarify. Thank you again for helping us improve this paper.
>
> Best regards,
>
> Submission 7385 Authors

---

### Author Response · Authors · 2025-11-20
**[Authors' Response] General Response to Shared Concerns: Novelty, Generality, and Scope**

We sincerely appreciate the reviewers' constructive feedback and their recognition of our work's conceptual novelty (Reviewers Das5, 99um and LJBz), extensive experimental scale (Reviewers Das5, 99um and LJBz), and the value of our findings (Reviewers Das5, mp9C and 99um). Below, we address three major shared concerns.

## 1. Novelty & Utility: Why PID instead of Ablation? (Addressing Reviewers Das5, dfu9, mp9C and LJBz)

**Critique:** Does PID offer value over simpler metrics like "accuracy delta" (Acc_full - Acc_text), or is it merely descriptive?

**Response:**

1. **Distribution-based vs. outcome-based:** Accuracy delta is a binary, label-dependent metric. PID is derived solely from the model's predicted distributions ($P(Y|X)$). The PID components $(R, U_1, U_2, S)$ are defined without reference to ground truth, and only afterward do we relate them to accuracy. This means correlations between, e.g., $S$ and performance are not baked into the metric’s definition.

2. **Disentangling mechanisms:** Accuracy delta conflates strategies. Gains might come from genuinely synergistic fusion $S$, from vision-only corrections $U_1$, or from language priors $U_2$ becoming more or less reliable. PID quantitatively separates these pathways.

3. **Additional experiment: comparing PID to accuracy deltas.** We correlated synergy with accuracy delta. On synergy-driven tasks, we found only a moderate linear correlation ($R^2_{\text{MMBench}} = 0.496$, $R^2_{\text{POPE}} = 0.554$), showing ablation only explains half of fusion behavior. Crucially, on knowledge-driven tasks, correlation is substantially weaker ($R^2_{\text{Reefknot}} = 0.149$, $R^2_{\text{PMC-VQA}} = 0.348$). In these cases, models can have similar accuracy deltas but very different levels of $S$ (e.g., Qwen2.5-VL vs. Gemma3). This demonstrates that PID captures differences in fusion strategies that are largely invisible to accuracy-based ablations.

**Summary:** PID acts as a diagnostic tool revealing *how* models solve tasks (e.g., Finding 4: scaling boosts fusion $S$, not just priors $U_2$), not just *whether* they solve them. We will clarify this distinction in the updated manuscript by explicitly contrasting PID-based analyses with simple accuracy deltas.

## 2. Generalizability of Regimes & Strategies (Addressing Reviewers dfu9, mp9C and LJBz)

**Critique:** The "synergy-driven vs. knowledge-driven" regimes may be tautological, and "family strategies" are inferred from only four datasets.

**Response:**

1. **Empirical regularity, Not definition:** We do not define tasks as "synergy-driven" by stipulating that their performance must correlate with S. Instead, we first measure the information spectrum $(R, U_1, U_2, S)$ across 26 models and then observe that datasets naturally separate into two regimes based on how models allocate information.

2. **Stability of strategies:** Crucially, family strategies are invariant across regimes. Fusion-centric families (e.g., Qwen2.5-VL) generate synergy even on knowledge-driven tasks; language-centric models (e.g., Gemma3) rely on priors even in synergy-driven contexts. This cross-regime consistency supports interpreting "fusion-centricity" and "language-centricity" as stable properties of families, not artifacts of any single dataset.

**Summary:** This confirms that our taxonomy is robust. We refined the main text to frame these as empirical patterns observed on various models and datasets.

## 3. Scope: Rationale for MC-VQA over Open Generation (Addressing Reviewers Das5, dfu9, 99um and LJBz)

**Critique:** MC-VQA restriction limits applicability to generative LVLMs.

**Response:** Our focus on MC-VQA is a deliberate choice to ensure that PID estimates are mathematically well-defined and empirically stable for modern LVLMs.

1. **Instability of Open-Ended Estimation:** Applying BATCH to open generation requires mapping high-dimensional outputs to a discrete $\mathcal{Y}$. Our preliminary experiments with manual pre-clustering like [1] and auxiliary linear probes revealed that these interventions introduce substantial hyperparameter sensitivity and estimation noise.

2. **Avoiding Distortions:** Introducing clustering or auxiliary layers acts as a "distortion," shifting the analysis from the LVLM's native representation to an artificially constructed space. This compromises the framework's ability to faithfully profile the base model's intrinsic information flow.

3. **Rigorous Proxy:** MC-VQA is a widely adopted format covering diverse domains (hallucination, domain knowledge) with a mathematically well-defined output space. This allows us to isolate the reasoning engine directly.

**Summary**: Our choice of MC-VQA prioritizes the fidelity and stability of PID estimation on LVLMs, while still covering diverse and practical behaviors. We will update Section 5 to detail this rationale.

[1]. Liang et al. Quantifying & modeling multimodal interactions: An information decomposition framework. NeurIPS 2023.

---

### Author Response · Authors · 2025-11-21
**[Author's Response] Update: Revised PDF Uploaded**

We have uploaded a revised version of the paper. The main updates are:

**Sections 3.2 & 3.3.1:** Clarified the rationale for focusing on MC-VQA and discrete outputs, emphasized that our PID framework provides a process-level and label-free decomposition of model behavior, and added a text-only intervention setting (image removal) that we later use as a sanity check.

**Section 4:** Refined the description of task regimes, added the intervention results relating synergy $S$ to the performance drop, and softened some statements to better reflect the empirical nature of our findings.

**Section 5:** Re-organized the conclusion, added an explicit limitations paragraph, and expanded the future-work discussion with potential applications of the framework, as a process-level diagnostic.

For the reviewers' convenience, the main newly added or substantially revised passages are marked in **blue** in this version; we will revert them to black in the final version.

We sincerely thank all reviewers for their thoughtful feedback and efforts in helping us refine this work once again.

---

### Meta-Review · Program_Chairs · 2026-01-22

**Summary:**

The reviewers raised three concerns:
1. Reviewers thought that the focus on multiple-choice VQA, as opposed to general VQA was limiting.
2. Reviewers questioned whether this complex analysis yielded any new insight beyond what might be discerned by simply looking at accuracy metrics.
3. Reviewers pointed out that the conclusions were primarily correlational and not causal in nature.

**Reviewer Concerns:**

The authors did a comprehensive rebuttal. Based on my understanding, this is how the concerns stand:
1. Multiple-choice VQA : the authors argue that this limitation is necessary for methodological rigor. This does not address the concern completely. They also point out that this restriction to multiple choice is common in the community. This point is well taken.
2. Comparison of this analysis vis-a-vis accuracy analysis: the authors demonstrate that simpler measures based on accuracy do not correlate with their findings. This in itself is not sufficient to suggest that the author's analysis is more useful. However, it does suggest that the paper's analysis provides signal that is complementary to established ways of performing diagnostic analysis.
3. Correlational vs causal: The authors ceded that the analysis was correlational; however, they did perform experiments with interventions suggested by the reviewers.

Overall, there are still outstanding concerns. However, the paper presents an analysis of VQA that is complementary to prior work. This may be a useful addition to the community. As such, I recommend accept.

**Reviewer Scores:**

Reviewer Das5 scored the paper as marginally above the bar. They wanted more fine-grained analysis, which was not provided. I do not think they would have changed their score.
Reviewer mp9C was concerned about why this complex metric is necessary. This point has been answered, they would have raised their score.
Reviewers dfu9 and 99um had concerns about the correlational vs causal nature of the study. This point has not been answered successfully, so they may have maintained their score.

---

### Decision · Program_Chairs · 2026-01-26

Accept (Poster)